



# Validation of a coupled $\delta^2H_{n\text{-alkane}}$-$\delta^{18}O_{sugar}$ paleohygrometer approach based on a climate chamber experiment

Johannes Hepp[a,b,1,*,#], Bruno Glaser[b], Dieter Juchelka[c], Christoph Mayr[d,e,2], Kazimierz Rozanski[f], Imke Kathrin Schäfer[g], Willibald Stichler[h], Mario Tuthorn[c,3], Roland Zech[g,i,4], Michael Zech[b,j,5,#]

[a]Chair of Geomorphology and BayCEER, University of Bayreuth, Universitätsstrasse 30, D-95440 Bayreuth, Germany
[b]Institute of Agronomy and Nutritional Sciences, Soil Biogeochemistry, Martin-Luther-University Halle-Wittenberg, Von-Seckendorff-Platz 3, D-06120 Halle (Saale), Germany
[c]Thermo Fisher Scientific, Hanna-Kunath-Str. 11, D-28199 Bremen, Germany
[d]Institute of Geography, Friedrich-Alexander-University Erlangen-Nürnberg, Wetterkreuz 15, D-91058 Erlangen, Germany
[e]GeoBio-Center & Earth and Environmental Sciences, Ludwig-Maximilian University Munich, Richard-Wagner-Str. 10, D-80333 München, Germany
[f]Faculty of Physics and Applied Computer Science, AGH University of Science and Technology, Al. Mickiewicza 30, PL-30-059 Kraków, Poland
[g]Institute of Geography and Oeschger Centre for Climate Research, University of Bern, Hallerstrasse 12, CH-3012 Bern, Switzerland
[h]Helmholtz Zentrum München, German Research Center for Environmental Health, Ingolstädter Landstrasse 1, D-85764 Neuherberg, Germany
[i]Institute of Geography, Chair of Physical Geography, Friedrich-Schiller University of Jena, Löbdergraben 32, D-07743 Jena, Germany
[j]Institute of Geography, Heisenberg Chair of Physical Geography with focus on paleoenvironmental research, Technical University of Dresden, Helmholtzstrasse 10, D-01062 Dresden, Germany

[*]corresponding author: johannes-hepp@gmx.de
[#]all other co-authors are listed alphabetically

[1]Present address: Chair of Geomorphology and BayCEER, University of Bayreuth, Universitätsstrasse 30, D-95440 Bayreuth, Germany
[2]Present address: Institute of Geography, Friedrich-Alexander-University Erlangen-Nürnberg, Wetterkreuz 15, D-91058 Erlangen, Germany
[3]Present address: Thermo Fisher Scientific, Hanna-Kunath-Str. 11, D-28199 Bremen, Germany
[4]Present address: Institute of Geography, Chair of Physical Geography, Friedrich-Schiller University of Jena, Löbdergraben 32, D-07743 Jena, Germany
[5]Present address: Institute of Geography, Heisenberg Chair of Physical Geography with focus on paleoenvironmental research, Technical University of Dresden, Helmholtzstrasse 10, D-01062 Dresden, Germany





**Keywords**
hydrogen stable isotopes, oxygen stable isotopes, hemicellulose sugars, leaf waxes, leaf water
enrichment, deuterium-excess, relative humidity

**Abstract**
The hydrogen isotopic composition of leaf wax-derived biomarkers, e.g. long chain $n$-alkanes ($\delta^2 H_{n\text{-}alkane}$),
is widely applied in paleoclimatology research. However, a direct reconstruction of the isotopic
composition of paleoprecipitation based on $\delta^2 H_{n\text{-}alkane}$ alone can be challenging due to the overprint of
the source water isotopic signal by leaf-water enrichment. The coupling of $\delta^2 H_{n\text{-}alkane}$ with $\delta^{18}O$ of
hemicellulose-derived sugars ($\delta^{18}O_{sugar}$) has the potential to disentangle this effect and additionally
allow relative humidity reconstructions. Here, we present $\delta^2 H_{n\text{-}alkane}$ as well as $\delta^{18}O_{sugar}$ results obtained
from leaves of the plant species *Eucalyptus globulus*, *Vicia faba* var. *minor* and *Brassica oleracea* var.
*medullosa*, which were grown under controlled conditions. We addressed the questions (i) do $\delta^2 H_{n\text{-}alkane}$
and $\delta^{18}O_{sugar}$ values allow precise reconstructions of leaf water isotope composition, (ii) how
accurately does the reconstructed leaf-water-isotope composition enables relative humidity (RH)
reconstruction in which the plants grew, and (iii) does the coupling of $\delta^2 H_{n\text{-}alkane}$ and $\delta^{18}O_{sugar}$ enable a
robust source water calculation?
For all investigated species, the alkane $n$-$C_{29}$ was most abundant and therefore used for compound-
specific $\delta^2 H$ measurements. For *Vicia faba*, additionally the $\delta^2 H$ values of $n$-$C_{31}$ could be evaluated
robustly. With regard to hemicellulose-derived monosaccharides, arabinose and xylose were most
abundant and their $\delta^{18}O$ values were therefore used to calculate weighted mean leaf $\delta^{18}O_{sugar}$ values.
Both $\delta^2 H_{n\text{-}alkane}$ and $\delta^{18}O_{sugar}$ yielded significant correlations with $\delta^2 H_{leaf\text{-}water}$ and $\delta^{18}O_{leaf\text{-}water}$,
respectively ($r^2 = 0.45$ and $0.85$, respectively; $p < 0.001$, $n = 24$). Mean fractionation factors between
biomarkers and leaf water were found to be -156‰ (ranging from -133 to -192‰) for $\varepsilon_{n\text{-}alkane/leaf\text{-}water}$
and +27.3‰ (ranging from +23.0 to 32.3‰) for $\varepsilon_{sugar/leaf\text{-}water}$, respectively. Using rearranged Craig-
Gordon equations with either $T_{air}$ or $T_{leaf}$ and measured $\delta^2 H_{leaf\text{-}water}$ or $\delta^{18}O_{leaf\text{-}water}$ as input variables, we
furthermore modeled climate chamber $RH_{air}$ and $RH_{leaf}$ values. Modelled $RH_{air}$ values, from the more
simplified Craig-Gordon model, turned out to be most accurate and correlate highly significantly with
measured $RH_{air}$ values ($R^2 = 0.84$, $p < 0.001$; RMSE = 6%). When combining $\delta^2 H_{leaf\text{-}water}$ and $\delta^{18}O_{leaf\text{-}water}$
values that are calculated from the alkane and sugar biomarkers instead of actually measured $\delta^2 H_{leaf\text{-}water}$
and $\delta^{18}O_{leaf\text{-}water}$ as input variables, the correlation of modelled $RH_{air}$ values with measured $RH_{air}$
values is getting worse, but is still highly significant with $R^2 = 0.54$, $p < 0.001$; RMSE = 10%. This
highlights the potential of the coupled $\delta^2 H_{n\text{-}alkane}$-$\delta^{18}O_{sugar}$ paleohygrometer approach for suitable
relative humidity reconstructions. Finally, the reconstructed source water isotope composition ($\delta^2 H_s$
and $\delta^{18}O_s$) as calculated from the coupled approach matches the source water in the climate chamber
experiment ($\delta^2 H_{tank\text{-}water}$ and $\delta^{18}O_{tank\text{-}water}$).





## 1 Introduction

Leaf-wax-derived biomarkers, such as long chain $n$-alkanes, and their stable hydrogen isotopic composition ($\delta^2H_{n\text{-alkane}}$) are widely applied in paleoclimatology research. Sedimentary $\delta^2H_{n\text{-alkane}}$ values correlate with $\delta^2H$ of precipitation (Huang et al., 2004; Mügler et al., 2008; Sachse et al., 2004; Sauer et al., 2001), confirming the high potential of $\delta^2H_{n\text{-alkane}}$ to establish $\delta^2H$ records of past precipitation (Hou et al., 2008; Rao et al., 2009; Sachse et al., 2012). However, the alteration of the isotopic signal as a result of the often unknown amount of leaf water enrichment caused by evapotranspiration can be several tens of per mil. This poses a challenge for accurate data interpretation (e.g. Zech et al., 2015), especially in respect of single proxy ($\delta^2H_{n\text{-alkane}}$)-based climate records. Apart from studies of sedimentary cellulose (Heyng et al., 2014; Wissel et al., 2008), the oxygen stable isotope composition of sugar biomarkers ($\delta^{18}O_{sugar}$) emerged as complementary paleoclimate proxy during the last decade (Hepp et al., 2015, 2017, Zech et al., 2013a, 2014a). The interpretation of the $\delta^{18}O_{sugar}$ values is comparable to those of $\delta^2H_{n\text{-alkane}}$. When sugars originate primarily from leaf biomass of higher terrestrial plants, they reflect the plant source water (which is often directly linked to the local precipitation) modified by evapotranspirative enrichment of the leaf water (Tuthorn et al., 2014; Zech et al., 2014a). The coupling of $\delta^2H_{n\text{-alkane}}$ with $\delta^{18}O_{sugar}$ values allows quantification of leaf-water isotopic enrichment and relative air humidity (Zech et al., 2013a). This approach was validated by Tuthorn et al. (2015) by applying it to topsoil samples along a climate transect in Argentina. Accordingly, the biomarker-derived relative air humidity values correlate significantly with actual air relative humidity from the respective study sites, highlighting the potential of the $\delta^2H_{n\text{-alkane}}$-$\delta^{18}O_{sugar}$ paleohygrometer approach.

The coupled approach is based on the observation that the isotope signature of precipitation ($\delta^2H_{precipitation}$ and $\delta^{18}O_{precipitation}$) typically plots on or adjacent to the global meteoric water line (GMWL), in a $\delta^2H$-$\delta^{18}O$ diagram. The GMWL is characterized by the equation $\delta^2H_{precipitation} = 8 \cdot \delta^{18}O_{precpitation} + 10$ (Dansgaard, 1964). In most cases, the local precipitation can be directly linked to the source water of plants, which is indeed soil water and eventually shallow groundwater. The isotopic composition of xylem water of plants readily reflects these sources (e.g. Dawson, 1993). However, leaf-derived biomarkers reflect the leaf water isotope composition, which is, unlike xylem water, prone to evapotranspiration (e.g. Barbour and Farquhar, 2000; Helliker and Ehleringer, 2002; Cernusak et al., 2003; Barbour et al., 2004; Cernusak et al., 2005; Feakins and Sessions, 2010; Kahmen et al., 2011; Sachse et al., 2012; Kahmen, Schefuß, et al., 2013; Tipple et al., 2013; Lehmann et al., 2017; Liu et al., 2017). During daytime, the leaf water is typically enriched in the heavy isotope compared to the source water because of the evapotranspirative enrichment through the stomata. Thereby, lighter water isotopes evaporate preferentially, which results in a deuterium-excess in the remaining water compared to the precipitation water ($d = \delta^2H - 8 \cdot \delta^{18}O$; according to Dansgaard, 1964). The degree of evapotranspirative enrichment is mainly controlled by the relative air humidity in the direct surrounding of the plant leaves (e.g. Cernusak et al., 2016). Although the biomarkers reflect the isotopic composition of leaf water, there is still a modification by the so-called biosynthetic fractionation during the biosynthesis, leading to an offset between leaf water and biomarker isotope composition. In case the biosynthetic fractionation is known and constant, there is a great potential that relative humidity can be derived from coupling $\delta^2H_{n\text{-alkane}}$ and $\delta^{18}O_{sugar}$ values.

The overall aim of this study is to evaluate the $\delta^2H_{n\text{-alkane}}$-$\delta^{18}O_{sugar}$ paleohygrometer approach by applying it to plant leaf material from three different plants grown in a climate chamber experiment under well controlled conditions. More specifically, we address the following questions:

(i)     which homologue and specific monosaccharide can be used to gain $\delta^2H_{n\text{-alkane}}$ and $\delta^{18}O_{sugar}$ results for the climate chamber plants leaf material, respectively,





| 111 | (ii) | how precisely do $\delta^2H_{n\text{-}alkane}$ and $\delta^{18}O_{sugar}$ values allow reconstructing $\delta^2H$ and $\delta^{18}O$ of leaf |
| 112 | | water, respectively, |
| 113 | (iii) | how accurately does the leaf-water-isotope composition reflect the relative humidity |
| 114 | | conditions, |
| 115 | (iv) | and does the coupling of $\delta^2H_{n\text{-}alkane}$ and $\delta^{18}O_{sugar}$ enable a robust source water calculation |
| 116 | | and how reliable are relative humidity reconstructions? |

## 2 Material and Methods

### 2.1 Climate chamber experiment

A phytotron experiment was conducted at the Helmholtz Zentrum München in Neuherberg during winter 2000/2001 (Mayr, 2002). Three different dicotyledon plant species (*Eucalyptus globulus*, *Vicia faba* var. *minor* and *Brassica oleracea* var. *medullosa*) were grown in eight chambers for 56 days under seven distinct climatic conditions (same conditions in chambers 4 and 8). Air temperature ($T_{air}$) were set to 14, 18, 24 and 30°C and and relative humidity ($RH_{air}$) to around 20, 30, 50, and 70% between 11 a.m. and 4 p.m. (Fig. 1B). During the rest of the day typical natural diurnal variations were aimed for (details in Mayr, 2002). Furthermore, uniform irrigation conditions were guaranteed via an automatic irrigation system, which was controlled by tensiometers installed in 9 cm substrate depth. The tank water used for irrigation was sampled periodically (intervals of one to three days) over the whole experiment and revealed only minor variability in its isotope composition ($\delta^{18}O_{tank\text{-}water}$ = -10.7 ± 0.3‰ standard deviation ($\sigma$); $\delta^2H_{tank\text{-}water}$ = -7 ± 1‰ $\sigma$). Once a week, soil water (via ceramic cups in 13 cm soil depth) and atmospheric water vapor (via dry ice condensation traps) was sampled ($\delta^2H_{soil\text{-}water}$, $\delta^{18}O_{soil\text{-}water}$ and $\delta^2H_{atmospheric\text{-}water\text{-}vapor}$, $\delta^{18}O_{atmospheric\text{-}water\text{-}vapor}$). Additionally, leaf temperatures ($T_{leaf}$) were derived from gas exchange measurements, at least once a week (Mayr, 2002).

In order to analyze stable hydrogen and oxygen isotopic composition of leaf ($\delta^2H_{leaf\text{-}water}$, $\delta^{18}O_{leaf\text{-}water}$) and stem water, the plants were harvested at the end of the experiment. The vacuum distillation method was used for the extraction of the plant water. It should be noted that stem water is a mixture between phloem and xylem water, while the latter should reflect the isotopic composition of the soil water. For simplification, stem water is referred to as xylem water in the following ($\delta^2H_{xylem\text{-}water}$, $\delta^{18}O_{xylem\text{-}water}$).

For more details about the experiment, the reader is referred to the original publication (Mayr, 2002).

### 2.2 Leaf biomarker extraction and compound-specific stable isotope analysis

A total of 24 leaf samples were prepared according to Schäfer et al. (2016) for compound specific $\delta^2H$ measurements of *n*-alkanes, at the Institute of Geography, Group of Biogeochemistry and Paleoclimate, University of Bern. Microwave extraction with 15 ml dichloromethane (DCM)/methanol (MeOH) 9:1 (v:v) at 100°C for 1 h was conducted. The resulting total lipid extract was purified and separated using aminopropyl-silica-gel (Supelco, 45 µm) pipette columns. The hydrocarbon fraction (containing *n*-alkanes) was eluted with *n*-hexane and cleaned via silver nitrate-coated silica gel pipettes (Supelco, 60-200 mesh) and zeolite (Geokleen Ltd.) columns. The $\delta^2H$ measurements of the highest concentrated *n*-alkanes (*n*-$C_{29}$ and *n*-$C_{31}$) were performed on a GC-$^2$H-pyrolysis-IRMS system, equipped with an Agilent 7890A gas chromatograph (GC) and IsoPrime 100 isotope-ratio-mass spectrometer (IRMS) coupled with a GC5 pyrolysis/combustion interface operating in pyrolysis modus with a Cr (ChromeHD) reactor at 1000°C. The compound-specific $\delta^2H$ values were calibrated against a standard alkane mix (*n*-$C_{27}$, *n*-$C_{29}$, *n*-$C_{33}$) with known isotope composition (A. Schimmelmann, University of Indiana), measured twice every six sample injections. Standard deviation of the triplicate





measurements were typically $\leq$ 5‰. The $H^{3+}$ factor stayed constant during the course of the
measurements.

Additionally, the leaf samples were dried and finely ground in preparation for $\delta^{18}O$ analysis of
hemicellulose-derived sugars (modified from Zech and Glaser, 2009) at the Institute of Agronomy and
Nutritional Sciences, Soil Biogeochemistry, Martin-Luther-University Halle-Wittenberg. The
hemicellulose sugars were hydrolytically extracted for 4 h at 105°C using 4M trifluoroacetic acid
(Amelung et al., 1996) and purified via XAD-7 and Dowex 50WX8 columns. Prior to the methylboronic-
acid (MBA) derivatization (4 mg of MBA in 400 μl dry pyridine for 1 h at 60°C), the cleaned sugars were
frozen and freeze-dried overnight (Knapp, 1979). Compound-specific $\delta^{18}O$ measurements were
performed on a Trace GC 2000 coupled to a Delta V Advantage IRMS via an $^{18}O$-pyrolysis reactor (GC
IsoLink) and a ConFlo IV interface (all devices from Thermo Fisher Scientific, Bremen, Germany). The
sample batches were measured along with embedded co-derivatized standard batches, which
contained arabinose, fucose, xylose, and rhamnose in different concentrations of known $\delta^{18}O$ value.
The $\delta^{18}O$ values of the standard sugars were determined via temperature conversion/elemental
analysis-IRMS coupling at the Institute of Plant Sciences, ETH Zurich, Switzerland (Zech and Glaser,
2009). This procedure allows corrections for possible amount dependencies (Zech and Glaser, 2009)
and ensures the "Principle of Identical Treatment" (Werner and Brand, 2001). Standard deviations for
the triplicate measurements were 0.9‰ and 2.2‰ (average over all investigated samples) for
arabinose and xylose, respectively. We focus on arabinose and xylose in this study because they were
(i) the dominant peaks in all chromatograms, and (ii) previously found to strongly predominate over
fucose (and rhamnose) in terrestrial plants, soils (Hepp et al., 2016).

All δ values are expressed in per mil as isotope ratios (R = $^{18}O/^{16}O$ or $^{2}H/^{1}H$) relative to the Vienna
Standard Mean Ocean Water (VSMOW) standard in the common delta notation
($\delta = R_{sample} - R_{standard}/R_{standard}$; e.g. Coplen, 2011).

**2.3 Framework for coupling $\delta^{2}H_{n\text{-alkane}}$ with $\delta^{18}O_{sugar}$ results**

**2.3.1 Deuterium-excess of leaf water and relative humidity**

The coupled approach is based on the observation that isotope composition of global precipitation
plots typically close to the GMWL ($\delta^{2}H_{precipitation} = 8 \cdot \delta^{18}O_{precipitation} + 10$; Dansgaard, 1964; Fig. 2). The
soil water and shallow groundwater, which acts as source water for plants, can often directly be related
to the local precipitation. However, especially during daytime leaf water is typically enriched compared
to the precipitation due to evapotranspiration through the stomata, therefore plotting right of the
GMWL (Fig. 2; e.g. Allison et al., 1985; Bariac et al., 1994; Walker and Brunel, 1990). The leaf water
reservoir at the evaporative sites is frequently assumed to be in isotope steady-state (Allison et al.,
1985; Bariac et al., 1994; Gat et al., 2007; Walker and Brunel, 1990), meaning that the isotope
composition of the transpired water vapor is in isotopic equilibrium with the source water utilized by
the plants during the transpiration process. The Craig-Gordon model (e.g. Flanagan et al., 1991; Roden
and Ehleringer, 1999) approximates the isotope processes in leaf water in δ terms (e.g. Barbour et al.,
196 2004):

$$\delta_e \approx \delta_s + \varepsilon^* + \varepsilon_k + (\delta_a - \delta_s - \varepsilon_k) \frac{e_a}{e_i}, \qquad\qquad \text{(Equation 1)}$$

where $\delta_e$, $\delta_s$ and $\delta_a$ are the hydrogen and oxygen isotopic compositions of leaf water at the evaporative
sites, source water and atmospheric water vapor, respectively. The equilibrium enrichment ($\varepsilon^*$) is
expressed as $(1 - 1/\alpha_{L/V}) \cdot 10^3$, where $\alpha_{L/V}$ is the equilibrium fractionation between liquid and vapor in



per mil. The kinetic fractionation parameter ($\varepsilon_k$) describes the water vapor diffusion from intracellular
air space through the stomata and the boundary layer into to the atmosphere, and $e_a/e_i$ is the ratio of
the atmospheric to intracellular vapor pressure.
In a $\delta^2$H-$\delta^{18}$O diagram, the isotope composition of the leaf water as well as the source water can be
described as deuterium-excess (d) values by using the equation of Dansgaard (1964), with $d = \delta^2H - 8 \cdot$
$\delta^{18}O$. This allows rewriting the Eq. 1, in which hydrogen and oxygen isotopes have to be handled in
separate equations, in one equation:

$$d_e \approx d_s + \left(\varepsilon_2^* - 8 \cdot \varepsilon_{18}^*\right) + \left(C_k^2 - 8 \cdot C_k^{18}\right) + \left[d_a - d_s - \left(C_k^2 - 8 \cdot C_k^{18}\right)\right] \cdot \frac{e_a}{e_i}, \qquad \text{(Equation 2)}$$

where $d_e$, $d_s$ and $d_a$ are the deuterium excess values of leaf water at the evaporative sites, source water
and atmospheric water vapor, respectively. The kinetic fractionation parameter ($\varepsilon_k$) is typically related
to stomatal and boundary layer resistances to water flux (Farquhar et al., 1989). We used the kinetic
enrichment factor ($C_k$) instead of $\varepsilon_k$ to be close to paleo studies were direct measurements of such a
plant physiological parameter are not available. The kinetic enrichment factor is derived from a more
generalized form of the Craig-Gordon model for describing the kinetic isotope enrichment for $^2$H and
$^{18}$O ($C_k^2$ and $C_k^{18}$, respectively) (Craig and Gordon, 1965; Gat and Bowser, 1991). If the plant source
water and the local atmospheric water vapor are in isotope equilibrium, the term $\delta_a - \delta_s$ in Eq. 1 can
be approximated by $-\varepsilon^*$. Thus, Eq. 2 can be reduced to:

$$d_e \approx d_s + \left(\varepsilon_2^* - 8 \cdot \varepsilon_{18}^* + C_k^2 - 8 \cdot C_k^{18}\right) \cdot \left(1 - \frac{e_a}{e_i}\right). \qquad \text{(Equation 3)}$$

The actual atmospheric vapor pressure ($e_a$) and the leaf vapor pressure ($e_i$) in kPa can be derived from
Eqs. 4 and 5 by using $T_{air}$ and $T_{leaf}$, respectively:

$$e_a = 0.61365 \cdot e^{[17.502 \cdot T_{air} / (T_{air} + 240.97)]} \cdot RH_{air} \qquad \text{(Equation 4)}$$

$$e_i = 0.61365 \cdot e^{[17.502 \cdot T_{air/leaf} / (T_{air/leaf} + 240.97)]}, \qquad \text{(Equation 5)}$$

where $e_a/e_i$ is the relative humidity calculated with the saturation vapor pressure when the leaf
temperature is used in the denominator rather than the air temperature (Eq. 5), ranging between 0
and 1. In order to increase the comparability to $RH_{air}$, the $e_a/e_i$ ratio calculated with $T_{leaf}$ in Eq. 5 can be
converted into $RH_{leaf}$ by multiplication with 100. When $T_{air}$ is used in Eq. 5, $e_a/e_i$ represents $RH_{air}$ (also
ranging between 0 and 1, representing 0 to 100% relative humidity when multiplying with 100). It
should be noted that the differences between measured $RH_{leaf}$ and $T_{leaf}$ with the respective air
parameters (RH, $T_{air}$) are not very pronounced in most cases (Mayr, 2002; Kahmen et al., 2011b),
revealing rather the same trends and magnitude (Fig. 1B).
With Eqs. 2 and 3, two equations are given to derive relative humidity values by rearranging them,
resulting in $RH_{air}$ and $RH_{leaf}$, respectively, by using either $T_{air}$ or $T_{leaf}$ for $\varepsilon^*$ (Eqs. 6 and 7):

$$RH_{leaf/air} \approx \frac{d_e - d_s - \left(\varepsilon_2^* - 8 \cdot \varepsilon_{18}^*\right) - \left(C_k^2 - 8 \cdot C_k^{18}\right)}{d_a - d_s - \left(C_k^2 - 8 \cdot C_k^{18}\right)}, \qquad \text{(Equation 6)}$$

$$RH_{leaf/air} \approx 1 - \frac{d_e - d_s}{\left(\varepsilon_2^* - 8 \cdot \varepsilon_{18}^* + C_k^2 - 8 \cdot C_k^{18}\right)}. \qquad \text{(Equation 7)}$$

Equilibrium fractionation parameters ($\varepsilon_2^*$ and $\varepsilon_{18}^*$) are derived from empirical equations of Horita and
Wesolowski (1994) by using either the climate chamber $T_{air}$ or $T_{leaf}$ values. The kinetic fractionation
parameters ($C_k^2$ and $C_k^{18}$) for $^2$H and $^{18}$O, respectively, are set to 25.1 and 28.5‰ according to Merlivat
(1978), who reported maximum values during the molecular diffusion process of water through a
stagnant boundary layer. It should be noted that $\varepsilon_k$ values of broadleaf trees and shrubs over broad



climatic conditions are well in the range with used $C_k^2$ and $C_k^{18}$ values, revealing 23.9 ± 0.9 and 26.7‰
± 1.0 for $\varepsilon_k^2$ and $\varepsilon_k^{18}$, respectively (derived from supplementary data of Cernusak et al., 2016).
If $\delta^2H_{leaf-water}$ and $\delta^{18}O_{leaf-water}$ can be reconstructed from the measured δ values of *n*-alkanes and sugars
biomarkers, this framework provides a powerful tool to establish relative humidity records from
sedimentary archives (Hepp et al., 2017; Zech et al., 2013a). To reconstruct the isotope composition of
leaf water it is assumed that fractionation factors of −160‰ for $^2H$ of alkanes *n*-$C_{29}$ and *n*-$C_{31}$ ($\varepsilon^2_{bio}$;
Sachse et al., 2012; Sessions et al., 1999), and +27‰ for $^{18}O$ of the hemicellulose-derived sugars
arabinose and xylose ($\varepsilon^{18}_{bio}$; Cernusak et al., 2003; Schmidt et al., 2001; Sternberg et al., 1986; Yakir
and DeNiro, 1990) can be applied:

$$\text{alkane-based } \delta^2H_{leaf-water} = (\delta^2H_{n\text{-}alkane} - \varepsilon^2_{bio})/(1 + \varepsilon^2_{bio}/1000) \qquad \text{(Equation 8)}$$

$$\text{sugar-based } \delta^{18}O_{leaf-water} = (\delta^{18}O_{sugar} - \varepsilon^{18}_{bio})/(1 + \varepsilon^{18}_{bio}/1000). \qquad \text{(Equation 9)}$$


**2.3.2 Isotope composition of plant source water**
In a $\delta^2H$-$\delta^{18}O$ diagram, the hydrogen and oxygen isotope composition of the plant source water ($\delta^2H_s$
and $\delta^{18}O_s$, respectively) can be assessed via the slope of the individual leaf water evapotranspiration
lines (LEL´s; Craig and Gordon, 1965; Gat and Bowser, 1991). Depending on the degree of
simplification, the LEL slope ($S_{LEL}$) can be derived from Eq. 10 (consistent to Eq. 2) and Eq. 11 (consistent
to Eq. 3):

$$S_{LEL} \approx \frac{\varepsilon_2^* + C_k^2 + (\delta_a^2 - \delta_s^2 - C_k^2) \cdot \frac{e_a}{e_i}}{\varepsilon_{18}^* + C_k^{18} + (\delta_a^{18} - \delta_s^{18} - C_k^{18}) \cdot \frac{e_a}{e_i}}, \qquad \text{(Equation 10)}$$

$$S_{LEL} \approx \frac{\varepsilon_2^* + C_k^2 \cdot \left(1 - \frac{e_a}{e_i}\right)}{\varepsilon_{18}^* + C_k^{18} \cdot \left(1 - \frac{e_a}{e_i}\right)} \approx \frac{\varepsilon_2^* + C_k^2}{\varepsilon_{18}^* + C_k^{18}}, \qquad \text{(Equation 11)}$$

where all parameters are defined as in section 2.3.1. The $\delta^2H_s$ and $\delta^{18}O_s$ values can then be calculated
for each leaf water data point via the intersect between the individual LEL´s with the GMWL. The model
results (from Eqs. 10 and 11) can be furthermore compared to the slope calculated by Eq. 12, using the
measured $\delta^2H_{leaf-water}$, $\delta^{18}O_{leaf-water}$ and $\delta^2H_{tank-water}$, $\delta^{18}O_{tank-water}$ values (Craig and Gordon, 1965; Gat and
Bowser, 1991).

$$S_{LEL} = \frac{\delta^2H_{leaf-water} - \delta^2H_{tank-water}}{\delta^{18}O_{leaf-water} - \delta^{18}O_{tank-water}} \qquad \text{(Equation 12)}$$


**2.4 Modeling and isotope fractionation calculations**
Relative humidity (Eq. 6), deuterium-excess values of leaf water ($d_e$, Eq. 2) and $S_{LEL}$ values (Eq. 10) were
modeled leading to less simplified results, because the measured $\delta_a$ values are used explicitly.
Equations 7, 3 and 11 were therefore used to obtain RH, $d_e$ and $S_{LEL}$ results, representing a more
simplified model approach because $\delta_a$ - $\delta_s$ are approximated by -$\varepsilon^*$. This model procedure allows
furthermore the comparison of scenarios based on air or leaf temperature ($T_{air}$ or $T_{leaf}$). In Eqs. 6 and
7, the reconstructed (biomarker-based) deuterium-excess$_{leaf-water}$ was used as additional input, as
gained from Eqs. 8 and 9. The modeled LEL slopes (Eqs. 10 and 11) were used to derive source water
isotope composition ($\delta^2H_s$, $\delta^{18}O_s$). In all equations presented in section 2.3 to gain the model results
(Eqs. 2 to 8), $\delta^2H_{atmospheric-water-voupor}$, $\delta^{18}O_{atmospheric-water-voupor}$ and $\delta^2H_{tank-water}$, $\delta^{18}O_{tank-water}$ were used for $\delta_a$
and $\delta_s$ (therefore also for $d_a$ and $d_s$). All other input parameters were set as described in section 2.3. In
order to provide an 1 σ range bracketing the modeled results ($d_e$, RH$_{air}$, RH$_{leaf}$, $S_{LEL}$, $\delta^2H_s$, $\delta^{18}O_s$), the





calculations were also run with values generated by subtracting/adding the individual σ to the average.
This procedure was also used to derive measured deuterium-excess$_{leaf-water}$ and S$_{LEL}$ uncertainties.
Model quality was overall assessed by calculating the coefficient of determination $\left[R^2 = 1 - \right.$
$\sum(\text{modeled - measured})^2 / \sum(\text{measured - measured mean})^2\left]\right.$ and the root mean square error
$\left[RMSE = \sqrt{\left(\frac{1}{n} \cdot \sum(\text{modeled - measured})^2\right)}\right]$. The R$^2$ is not equal to the r$^2$, which provides here the
fraction of variance explained by a linear regression between a dependent (y) and an explanatory
variable $[r^2 = 1 - \sum(y - \text{fitted y})^2/\sum(y - \text{mean y})^2]$ (R Core Team, 2015).

The fractionation between the measured leaf biomarkers and leaf water can be described by the
following equations (Eq. 10 and 11; e.g. Coplen, 2011):

$$\varepsilon_{n\text{-alkane/leaf-water}} = (\delta^2 H_{n\text{-alkane}} - \delta^2 H_{leaf\text{-water}}) / (1 + \delta^2 H_{leaf\text{-water}}/1000) \qquad \text{(Equation 13)}$$

$$\varepsilon_{sugar/leaf\text{-water}} = (\delta^{18}O_{sugar} + \delta^{18}O_{leaf\text{-water}}) / (1 + \delta^{18}O_{leaf\text{-water}}/1000). \qquad \text{(Equation 14)}$$

For Eqs. 8 and 9 (biomarker-based leaf water reconstruction) as well as for Eqs. 13 and 14, the 1 σ
range were calculated by subtracting/adding the individual σ, analogous to the modeling results.

All calculations and statistical analysis were realized in R (version 3.2.2; R Core Team, 2015).

**3 Results and Discussion**
**3.1 Compound-specific isotope results of leaf wax-derived *n*-alkanes and hemicellulose-**
**derived sugars**
All investigated leaf material showed a dominance of C$_{29}$ *n*-alkanes. The dominance of *n*-C$_{29}$ in *Brassica*
*oleracea* and *Eucalyptus globulus* was also reported by Ali et al. (2005) and Herbin and Robins (1968).
*Vicia faba* leaf samples additionally revealed a high abundance of C$_{31}$ *n*-alkanes. This agrees with results
from Maffei (1996) and enables a robust determination of compound-specific δ$^2$H values for C$_{29}$ and
C$_{31}$. The δ$^2$H$_{n\text{-alkane}}$ values of *Vicia faba* are therefore calculated as weighted mean.
The top of Fig. 1A illustrates the δ$^2$H$_{n\text{-alkane}}$ results along with isotopic data for leaf, xylem and soil water
(the latter were originally published in Mayr 2002). In addition the climate chamber conditions (RH$_{air}$,
RH$_{leaf}$, T$_{air}$ and T$_{leaf}$) are displayed (all from Mayr, 2002; Fig. 1B). For more details about the (plant) water
isotope results, climate chamber conditions as well as not shown plant physiological properties the
reader is referred to Mayr (2002). The δ$^2$H$_{n\text{-alkane}}$ values range from -213 to -144‰ over all plant species.
As revealed by overlapping notches in the respective boxplots, no statistically significant differences in
the median values between the three plant species can be described (Fig. S1A; McGill et al., 1978). Fig.
1A moreover shows that δ$^2$H$_{n\text{-alkane}}$ values range largest for *Eucalyptus globulus* compared to the other
two plants. However, the low number of samples per plant species prohibits a robust interpretation.

(Fig. 1)

The investigated leaf samples yielded substantially higher amounts of arabinose and xylose compared
to fucose and rhamnose. This is in agreement with sugar patterns reported for higher plants (D'Souza
et al., 2005; Hepp et al., 2016; Jia et al., 2008; Prietzel et al., 2013; Zech et al., 2012, 2014a) and
hampers a robust data evaluation of fucose and rhamnose. The δ$^{18}$O values of the investigated
pentoses arabinose and xylose range from 30 to 47‰ and 30 to 50‰, respectively, and are shown





along with isotopic data for leaf, xylem and soil water (Mayr 2002) in the bottom of Fig. 1A. No
considerable difference in the $\delta^{18}O$ values of arabinose and xylose can be seen in the $\delta^{18}O$ pentose
data. This is in line with findings from Zech and Glaser (2009), Zech et al. (2012), Zech et al. (2013b)
and Zech et al. (2014b) but contradicting with slightly more positive $\delta^{18}O_{arabinose}$ values compared to
$\delta^{18}O_{xylose}$ values reported by Zech et al. (2013a) and Tuthorn et al. (2014). Overall, the two sugars
display very similar results (Fig. 1; $r^2 = 0.7$, $p < 0.001$, $n = 24$). The $\delta^{18}O$ values of arabinose and xylose
can therefore be combined as a weighted mean (as $\delta^{18}O_{sugar}$ values) for further data interpretation.
The $\delta^{18}O_{sugar}$ values are not significantly different between the three investigated plant species.
The compound-specific isotope results of leaf hemicellulose-derived sugars and leaf wax-derived *n*-
alkanes can be compared with leaf, xylem, soil and tank water (compare Fig. 1A and Fig. 2). This
comparison reveals that soil and xylem water plot close to the tank water, whereas leaf water shows
a clear evapotranspirative enrichment. This enrichment strongly differs between the climate
chambers, depending mainly on T and RH conditions. The biomarker results furthermore follow the
leaf water with a certain offset ($\varepsilon_{bio}$).
(Fig. 2)

### 3.2 Do *n*-alkane and sugar biomarkers reflect the isotope composition of leaf water?

The $\delta^2H_{n\text{-}alkane}$ dataset reveals a significant correlation with $\delta^2H_{leaf\text{-}water}$ of 0.45 ($r^2$) using all plant species
with $p < 0.001$ (Fig. 3A). A slope of 1.1 and an intercept of -152‰ furthermore characterize the
relationship. It seems that each plant type shows a different $\delta^2H_{n\text{-}alkane}$ to $\delta^2H_{leaf\text{-}water}$ relation, with the
highest slope for *Vicia faba* and the lowest for *Brassica oleracea*. However, we argue that the number
of replicates for each plant species is simply too low to interpret this finding robustly. A highly
significant correlation is also observed for the correlation between $\delta^{18}O_{sugar}$ and $\delta^{18}O_{leaf\text{-}water}$ ($r^2 = 0.84$,
$p < 0.001$; Fig. 3B). The regression reveals a slope of 0.74 and an intercept of 30.7‰.
(Fig. 3)
Since it is well known that measured leaf water is not always equal to the specific water pool in which
the *n*-alkanes are biosynthesized (e.g. Tipple et al., 2015), the correlation reveals a rather low $r^2$ (Fig.
3A). Furthermore, NADPH is acting also as hydrogen source during *n*-alkane biosynthesis, which is
clearly more negative than the biosynthetic water pool (Schmidt et al., 2003), further contributing to
a weakening of the $\delta^2H_{n\text{-}alkane}$ to $\delta^2H_{leaf\text{-}water}$ relationship. The correlation between the deuterium
contents of leaf wax *n*-alkanes and leaf water presented here is still well in range with the literature.
Feakins and Sessions (2010) presented *n*-alkane ($C_{29}$ and $C_{31}$) and leaf water $\delta^2H$ data from typical plant
species (excluding grasses) along a southern California aridity gradient, revealing that only $\delta^2H$ of *n*-$C_{29}$
is significantly correlated with leaf water ($r^2 = 0.24$, $p < 0.1$, $n = 16$; based on the associated
supplementary data). Another field dataset from the temperate forest at Brown's Lake Bog, Ohio, USA
revealed significant correlations between $\delta^2H$ of *n*-$C_{29}$ and *n*-$C_{31}$ with leaf water of the species *Prunus
serotina*, *Acer saccharinum*, *Quercus rubra*, *Quercus alba*, and *Ulmus americana* ($r^2 = 0.49$, $p < 0.001$,
$n = 38$; $r^2 = 0.59$, $p < 0.001$, $n = 29$; as derived form the supplement material of Freimuth et al., 2017).
Data from a controlled climate chamber experiment using two tree species show a highly significant
relationship between leaf wax *n*-alkanes $\delta^2H$ and leaf water (with $C_{31}$ of *Betula occidentalis* and $C_{29}$ of
*Populus fremontii*; $r^2 = 0.96$, $p < 0.001$, $n = 24$; derived from supplementary data of Tipple et al., 2015).
It is conformed that leaf wax *n*-alkanes of dicotyledonous plants largely incorporate the leaf water




isotope signal, while in monocotyledonous plants (e.g. grasses) the *n*-alkanes are more strongly
affected by the source water due to the leaf growth at the intercalary meristem (Kahmen et al., 2013).
The observed slope of the $\delta^{18}O_{sugar}$ to $\delta^{18}O_{leaf-water}$ relationship (Fig. 3B) could serve as indicator for a
leaf water (enrichment) signal transfer damping of approximately 26%. The theory behind the signal
damping is adopted from the cellulose research (e.g. Barbour and Farquhar, 2000). Barbour and
Farquhar (2000) related the extent of the signal damping to the proportion of unenriched source
water, which contribute to the local synthesis water pool and to the proportion of exchangeable
oxygen during cellulose synthesis. Here calculated damping factor would be well in the range of values
reported for cellulose synthesis in *Gossypium hirsutum* leaves (between 35 and 38%; Barbour and
Farquhar, 2000), for *Eucalyptus globulus* leaf samples (38%; Cernusak et al., 2005) and for five $C_3$ and
$C_4$ grasses (25%; Helliker and Ehleringer, 2002). Recently Cheesman and Cernusak (2017) provided
damping factors for leaf cellulose synthesis based on plant data grown under same conditions at
Jerusalem Botanical Gardens published by Wang et al. (1998), ranging between 4 and 100% with a
mean of 49%, revealing large variations among and between ecological groups (namely conifers,
deciduous, evergreen and shrubs). A large range of damping factors associated with leaf cellulose was
also reported by Song et al. (2014) for *Ricinus communis* grown under controlled conditions. A common
disadvantage of the above-mentioned studies is the absence of direct measurements of the proportion
of depleted source water contribution to the local synthesis water (as noticed by Liu et al., 2017), which
largely contribute to the extent of the damping factor (Barbour and Farquhar, 2000). However, when
transferring cellulose results to pentoses, such as hemicellulose-derived arabinose and xylose, it should
be noted that they are biosynthesized via decarboxylation of the carbon at position six (C6) from
glucose (Altermatt and Neish, 1956; Burget et al., 2003; Harper and Bar-Peled, 2002). Waterhouse et
al. (2013) showed that the oxygen atoms at C6 position in glucose moieties, used for heterotrophic
cellulose synthesis, are strongly affected by the exchange with local water (up to 80%). Based on these
findings, it can be suggested that the influence of the non-enriched source water during the synthesis
of leaf hemicelluloses is rather small.

**3.3 Fractionation factors between biomarkers and leaf water**
In order to explore possible species-specific effects on the fractionation between the biomarkers and
the leaf water, boxplots of the individual plant species of $\varepsilon_{n\text{-alkane/leaf-water}}$ and $\varepsilon_{sugar/leaf-water}$ values are
shown in Fig. 4. Median $\varepsilon_{n\text{-alkane/leaf-water}}$ values are -155‰ for *Brassica oleracea*, -164‰ for *Eucalyptus*
*globulus* and -149‰ for *Vicia faba* (Fig. 4A), with an overall mean value of -156‰ (ranging from -133
to -192‰). Median $\varepsilon_{sugar/leaf-water}$ values of +27.0‰ for *Brassica oleracea*, +26.6‰ for *Eucalyptus*
*globulus*, +26.8‰ for *Vicia faba* are shown in Fig. 4B. The overall $\varepsilon_{sugar/leaf-water}$ average value of the
three investigated species is +27.3‰ (ranging from +23.0 to +32.3‰). In both plots, no difference
between the individual species seems to be observable.

(Fig. 4)

The boxplots of $\varepsilon_{n\text{-alkane/leaf-water}}$ reveal that the median of the three investigated plant species can be
statistically not distinguished, due to overlapping notches (Fig. 4A). It should be noted that due to the
low sample number from each species, the 95% confidence interval is larger than the interquartile
range in some cases. However, it seems that at least small species-specific differences cannot be ruled
out. Our $\varepsilon_{n\text{-alkane/leaf-water}}$ values resemble well the data from a laboratory study (Kahmen et al., 2011),
reporting a median value of -162‰ for *n*-$C_{25}$, *n*-$C_{27}$ and *n*-$C_{29}$ of *Populus trichocarpa*. Furthermore, they
are well comparable to climate chamber data of *Betula occidentalis* (*n*-$C_{31}$) and *Populus fremontii* (*n*-





C$_{29}$) from Tipple et al. (2015), reporting a median $\varepsilon_{\textit{n}\text{-alkane/leaf-water}}$ value of -155‰. In addition, field
experiments reveal similar median values of -151‰ (for $\textit{n}$-C$_{29}$) and -142‰ (for $\textit{n}$-C$_{31}$) from typical plant
species (excluding grasses) from southern California (Feakins and Sessions, 2010) and -144‰ (for $\textit{n}$-
C$_{29}$, of the species *Prunus serotina*, *Acer saccharinum*, *Quercus rubra*, *Quercus alba* and *Ulmus*
*americana*) from the temperate forest at Brown's Lake Bog, Ohio, USA. The large range in $\varepsilon_{\text{xylem-water/leaf-}}$
$_{\text{water}}$ values from our study (-192 to -133‰) is also obvious in the respective laboratory and field studies
(-198 to -115‰, derived from $\textit{n}$-C$_{29}$ and $\textit{n}$-C$_{31}$ data from Feakins and Sessions, 2010; Kahmen et al.,
2011a; Tipple et al., 2015; Freimuth et al., 2017). This could point to a specific water pool being used
rather than bulk leaf water during biosynthesis (Sachse et al., 2012; Schmidt et al., 2003). In more
detail, alkane synthesis takes place by modifying/expanding fatty acids in the cytosol, while fatty acids
are synthesized in the chloroplasts (Schmidt et al., 2003). Thus, the cytosol as well as chloroplast water
is one hydrogen source. However hydrogen can additionally be added to the alkanes and fatty acids
by NADPH which originates from different sources (photosynthesis and pentose phosphate cycle,
Schmidt et al., 2003). It is therefore challenging to measure directly the water pool in which the alkanes
are biosynthesized (Tipple et al., 2015). Moreover, biosynthetic and metabolic pathways in general
(Kahmen et al., 2013; Sessions et al., 1999; Zhang et al., 2009), the carbon and energy metabolism of
plants more specifically (Cormier et al., 2018) and the number of carbon atoms of the $\textit{n}$-alkane chains
(Zhou et al., 2010) may have an influence on the fractionation. Our $\varepsilon_{\textit{n}\text{-alkane/leaf-water}}$ values correlate with
T$_{\text{air}}$ (Fig. S2A), whereas the correlation with RH$_{\text{air}}$ (Fig. S2B) is not significant. This could point to a
relationship between $\varepsilon_{\text{xylem-water/leaf-water}}$ and plant physiological processes (affecting various plants
differently).
The $\varepsilon_{\text{sugar/leaf-water}}$ values (Fig. 4B) do not correlate significantly with T$_{\text{air}}$, but significantly with RH$_{\text{air}}$ (Fig.
S2C and D). A temperature dependence of the $\varepsilon_{\text{sugar/leaf-water}}$ is not supported by this experiment, in
contrast to results from Sternberg and Ellsworth (2011), where a temperature effect on oxygen
fractionation during heterotrophic cellulose biosynthesis is observed. The here observed fractionation
between hemicellulose-derived sugars and leaf water, with regard to $\varepsilon_{\text{sugar/leaf-water}}$ values, is well in
range with values reported for sucrose (exported from photosynthesizing leaves) and leaf water, which
was shown to be +27‰ (Cernusak et al., 2003). Also the cellulose biosynthesis is associated with an
enrichment of around +27‰ compared to the synthesis water as shown in growth experiments
(Sternberg et al., 1986; Yakir and DeNiro, 1990). The relatively uniform fractionation is explained via
the isotope exchange between the carbonyl oxygens of the organic molecules and the surrounding
water (cf. Schmidt et al., 2001). This equilibrium fractionation effect was indeed described earlier by
the reversible hydration reaction of acetone in water by Sternberg and DeNiro (1983) to be +28, +28
and +26‰ at 15, 25 and 35°C, respectively. However, the observed range of approximately 9‰ (Fig.
4B) could indicate that partially more than the oxygen equilibrium fractionation between organic
molecules and medium water have to be considered. Presumably, isotopic as well as sucrose synthesis
gradients within the leaf have to be taken into account when interpreting leaf sugar oxygen isotopic
compositions and their correlation to leaf water (Lehmann et al., 2017). Lehmann et al. (2017) reported
on a fractionation between sucrose and leaf water of +33.1‰. Based on this they proposed a
conceptual scheme how such gradients can lead to discrepancies between the isotopic composition of
the bulk leaf water and the synthesis water, while the latter is incorporated into the carbohydrates,
and thus fractionation determination based on bulk leaf water can exceed the common average of
+27‰. Also Mayr et al. (2015) found a fractionation between aquatic cellulose $\delta^{18}$O and lake water
larger than this value of around +29‰.

**3.4 Strong control of relative humidity over deuterium-excess of leaf water**





The correlations between leaf water-based and measured $RH_{air}$ or $RH_{leaf}$ as well as modeled $d_e$ and
measured deuterium-excess$_{leaf-water}$ are illustrated in Fig. 5A, B, D and E. Furthermore, modeled LEL
slopes are compared to measured LEL slopes in Fig. 5C and F. In red, the results of the less simplified
models are displayed (Eqs. 6, 2 and 10), in black the results of the more simplified models are shown
(Eqs. 7, 3 and 11).
(Fig. 5)
Evidence for the strong control of relative humidity on deuterium-excess of leaf water comes from
multivariate regression analysis between the measured deuterium-excess$_{leaf-water}$ values versus $RH_{air}$,
$RH_{leaf}$ and $T_{air}$, $T_{leaf}$. The results reveal that the deuterium-excess$_{leaf-water}$ significantly correlates with $RH_{air}$
of the climate chambers ($p < 0.001$), with an $r^2$ of 0.92. When $RH_{leaf}$ and $T_{leaf}$ values are used, the $r^2$ is
0.84 and deuterium-excess$_{leaf-water}$ correlates significantly with $RH_{leaf}$ ($p < 0.001$). The strong control of
relative humidity on deuterium-excess of leaf water is furthermore supported by the significant
correlations between calculated versus measured $RH_{air}$ values (Fig. 5A), regardless of whether the Eq.
6 or 7 were used (representing a lower and higher degree of simplification). This is in line with the
strong correlation between modeled $d_e$ based on $T_{air}$ and measured deuterium-excess$_{leaf-water}$ values
(Fig. 5B). When modeled $RH_{leaf}$ values are compared to the measured ones, the correlation is less
strong compared to $RH_{air}$ (Fig. 5D vs. 5A), represented by lower $R^2$ and higher RMSE values. Clearly
more data points are lying above the 1:1 line with regard to $RH_{leaf}$, compared to $RH_{air}$. On the same
basis, the $T_{leaf}$-based $d_e$ shows a weaker correlation to the measured values than the $T_{air}$-based $d_e$ (Fig.
5E vs. 5B). The generally better model performance when $T_{air}$ is used (in contrast to $T_{leaf}$) could point
to the fact that $T_{leaf}$ does not well represent the actual conditions in the leaves. For the correlation
between modeled and measured $RH_{leaf}$ this means that the measured $RH_{leaf}$ values do not reflect the
real conditions because measured $RH_{leaf}$ is calculated via $e_i/e_a$ *100 with $T_{leaf}$ as input for the $e_a$ equation
(see section 2.3). In fact, the RH model results do not differ from each other and can be well compared
to the measured $RH_{air}$, while the measured $RH_{leaf}$ values reveal an average offset of approximately 9%
with regard to the median values (Figure S3A). This can be explained by the small difference in ε*
calculated either with $T_{leaf}$ or $T_{air}$. Moreover, when $T_{leaf}$ values are used to model $d_e$, the match to $T_{air}$-
based $d_e$ and measured deuterium-excess$_{leaf-water}$ values is weaker (Fig. 5B vs. E; Fig. S3B). This offset is
caused by higher $T_{leaf}$ values (compared to $T_{air}$; Fig. 1), which are leading to more negative modeled $d_e$
values.
Overall, the modeled $d_e$ values show a high agreement with measured deuterium-excess of leaf water
despite without being too positive, which can be expected from the literature. This is because bulk leaf
is less enriched than the leaf water at the evaporative sites, which is however, the output of the Craig-
Gordon-based leaf water enrichment model (e.g. Allison et al., 1985; Barbour et al., 2004; Cernusak et
al., 2016; section 2.3). Especially under low relative humidity conditions, the discrepancy between
Craig-Gordon model results and the measured values is shown to be more pronounced, associated
with higher transpiration fluxes and higher isotope heterogeneity within the leaf water due to a non-
uniform closure of the stomata (Flanagan et al., 1991; Santrucek et al., 2007). An overestimation of the
Craig-Gordon models can hardly be observed here (Fig. 5B and 5E). However, based on the accepted
leaf water enrichment theory (e.g. Cernusak et al., 2016), higher transpiration rates (e.g. under low
humidity conditions) should still lead to a larger discrepancy between Craig-Gordon modelled and
measured leaf water, because the back diffusion of enriched leaf water from the evaporative sites
should get lower the higher the transpiration flux is. Why there is no difference between modeled and



measured deuterium-excess of leaf water in here presented climate chamber experiment is not
comprehensible.
The simplified model variants show generally a better correspondence between calculated and
measured deuterium-excess of leaf water, based on $R^2$ and RMSE, than the less simplified models. This
does not seem to be related to the slope of the LEL because it can only be linked to the measured
values based on the less simplified models (Fig. 5C and 5F). The simplified air and leaf temperature
based slopes average at 2.7 and 2.6, respectively, with a common range between 2.5 and 2.8. The
average is well in agreement with the mean measured $S_{LEL}$ of 2.9. In addition, a regression through the
tank water and all leaf water points reveals a slope of 2.7 (± 0.02, based on subtracting/adding the
individual σ; $r^2 = 0.98$, n = 48, p < 0.001). This could be the reason why the more simplified models are
still more accurate, despite the less simplified models do not reflect well the range of the measured
$S_{LEL}$, which vary between 2.4 and 3.8. Much better matches are found for the less simplified LEL slopes
($T_{air}$ based: 2.6 and 3.8, $T_{leaf}$ based: 2.5 and 3.5; Fig. 5C and 5F). Indeed the measured as well as the
calculated $S_{LEL}$ depend on the $e_a/e_i$ ratio (hence $RH_{leaf}$ and $RH_{air}$ regarding $T_{leaf}$ or $T_{air}$ is used for
calculations, respectively) and on $δ_a - δ_s$, in line with the theory and literature (see section 2.3; e.g.
Allison et al., 1985). The higher accuracy of the simpler models would therefore imply that the $S_{LEL}$
depend only on equilibrium and kinetic fractionation parameters for both isotopes, which would valid
for isotope equilibrium conditions between the tank water (the water source of the plants) and the
atmospheric water vapor, allowing the usage of the unambiguous approximation $δ_a - δ_s = -ε^*$. Indeed,
close-to equilibrium conditions between the tank water and the atmospheric water vapor are observed
for the climate chambers 4 to 6 and 8, while the others are characterized by a slight disequilibrium
conditions. However, the degree of uncertainty seems to be higher when using $d_a$ values, by the
probably inadequate representation of the measured $δ^2H_{atmospheric-water-vapor}$ and $δ^{18}O_{atmospheric-water-vapor}$
with the actual conditions influencing the plants in the climate chamber, leading to a generally better
performance of the more simplified model variants.

**3.5 Coupling $δ^2H_{n-alkane}$ and $δ^{18}O_{sugar}$ – Potential and limitations**
One of the advantages of the proposed coupled $δ^2H_{n-alkane}$-$δ^{18}O_{sugar}$ approach is a more robust
reconstruction of the isotope composition of the source water, which can often be directly linked to
the local precipitation signal (Hepp et al., 2015, 2017; Tuthorn et al., 2015; Zech et al., 2013a).
Therefore, Fig. 6 shows boxplots for measured leaf water, biomarker-based (reconstructed) leaf water,
measured source water (tank water; see section 2.1), biomarker-based source water (using
reconstructed leaf water as origin for the LEL´s) and leaf-water-based source water values (using
measured leaf water as origin for the LEL´s). Source water isotope compositions were calculated via
the slopes of the LEL´s and the GMWL. The numbers (1-4) mark the available scenarios for source water
reconstruction (see section 2.4): 1) $S_{LEL}$ calculated with the more simplified Eq. 11 with $T_{air}$, 2) as 1 but
with $T_{leaf}$, 3) $S_{LEL}$ calculated with Eq. 10 with $T_{air}$, 4) as 3 but with $T_{leaf}$. Fig. 6 clearly shows that the $n$-
alkane and sugar biomarkers reflect leaf water rather than tank water used for irrigation. For $δ^2H$,
neither the range nor the median of the $δ^2H_{leaf-water}$ are well captured by the alkane-based leaf water
values. However, the overlapping notches do not support a statistical difference in the median values
(Fig. 6A). The medians are still on average 13‰ more positive than the measured $δ^2H_{tank-water}$. A higher
agreement between measured and modeled values is observed from leaf water-based $δ^2H_s$ compared
to $δ^2H_{tank-water}$. The average offset is reduced to 2‰ and the range is reduced by approximately 70‰,
compared to the biomarker-based reconstruction. Besides the more simplified leaf water-based $δ^2H_s$
using $T_{leaf}$ for calculating $ε^*$ (scenario 2 in Fig. 6A), no statistical significant difference can be seen
between the leaf water-based $δ^2H_s$ and the $δ^2H_{tank-water}$, with regard to the overlapping notches.



(Fig. 6)

For $\delta^{18}O$, the sugar-based leaf water values are in agreement with the measured ones with regard to
the median values, as supported by the largely overlapping notches (Fig. 6B). The range of the
reconstructed leaf water is in the order of 6‰ smaller than for the measured $\delta^{18}O_{leaf-water}$ dataset. All
reconstructed $\delta^{18}O_s$ values, regardless whether they are biomarker- or leaf water-based, are
comparable to the measured $\delta^{18}O_{tank-water}$. While the biomarker-based datasets depict an average
offset of 2‰, the leaf water-based values only differ by 0.3‰ from the tank water $\delta^{18}O$ values,
referring to the medians. As for $\delta^2H$, the same leaf water-based $\delta^{18}O_s$ scenario (more simplified leaf
water-based model using $T_{leaf}$ for calculating $\varepsilon^*$, scenario 2 in Fig. 6B) do not show overlapping notches
with $\delta^{18}O_{tank-water}$, while the other leaf water-based source water reconstructions do. In addition, the
range in the leaf water-based $\delta^{18}O_{source-water}$ values is considerable smaller than for the biomarker-based
once (9‰ reduction). The overall larger range in modeled $\delta^2H_s$ and $\delta^{18}O_s$ compared to measured
$\delta^2H_{tank-water}$ and $\delta^{18}O_{tank-water}$ can related to uncertainties in $S_{LEL}$ modeling (see equations in section
2.3.2). Bariac et al. (1994) mentioned that they found no agreement between the intersect of modeled
LEL´s with the GMWL and the plant source water. Allison et al. (1985) explained such results with
changing environmental conditions, leading to various LEL´s with a locus line not necessarily passing
the $\delta^2H_s$ and $\delta^{18}O_s$ data point, in a system that approaches rapidly new steady-state conditions.

Finally, the alkane and sugar-based leaf water values were used to reconstruct $RH_{air}$ and $RH_{leaf}$. While
the measured $RH_{air}$ is well captured by the biomarker-based air relative humidity values ($R^2$ = 0.54 and
0.48 for the more and less simplified models, respectively, Fig. 7A), the correlations are weak between
the reconstructed leaf relative humidity values and the measured $RH_{leaf}$ ($R^2$ = 0.09 and -0.04 for the
more and less simplified models, respectively, Fig. 7B). The measured $RH_{air}$ is reconstructed most
accurate by the biomarker-based air relative humidity values (Fig. 7A). As for leaf water-based RH
reconstructions, a difference between biomarker-based $RH_{air}$ and $RH_{leaf}$ is observed (compare Fig. 7B
with 7A). This can be explained by the small difference between $T_{leaf}$ and $T_{air}$, used for $\varepsilon^*$ calculations
in the respective equations. The better performance of the more simplified models compared to the
less simplified ones, in general, and the fact that $T_{air}$ seems to be the better model input compared to
$T_{leaf}$, more specifically, can be explained as for the leaf water-based application (see section 3.3). The
$T_{leaf}$ as well as the measured $\delta^2H_{atmospheric-water-vapor}$ and $\delta^{18}O_{atmospheric-water-vapor}$ values seem to be less
representative for the conditions affecting the climate chamber plant leaves.
(Fig. 7)

Overall, a lower coefficient of determination of the biomarker-based model results compared to the
leaf water-based reconstructions (compare Fig. 5A and D with Fig. 7A and B) is observed. This can be
attributed to the uncertainties in leaf water reconstructed using $\delta^2H_{n-alkane}$ and $\delta^{18}O_{sugar}$ datasets as
discussed in section 3.2. The limitations regarding deuterium arose from the rather weak relationship
between the $\delta^2H$ of the $n$-alkanes and the leaf water, probably linked with the large range in the
fractionation between $n$-alkanes and leaf water ($\varepsilon^2_{n-alkane/leaf-water}$). The applied equation to
reconstructed $\delta^2H_{leaf-water}$ by using $\delta^2H_{n-alkane}$ and a constant biosynthetic fractionation of -160‰ (Eq.
13) was considered to be suitable (Sachse et al., 2012; Sessions et al., 1999), but introduce also some
uncertainty for the final relative humidity reconstruction. With regard to oxygen, the relatively large
variations in $\varepsilon_{sugar/leaf-water}$ of 9‰ have to be considered (Fig. 4B), because in the $\delta^{18}O_{leaf-water}$



reconstructions a fixed value of +27‰ is used (Eq. 14). Such a uniform biosynthetic fractionation is an
approximation which may not always be fulfilled, as shown in the literature (e.g. Sternberg and
Ellsworth, 2011; Lehmann et al., 2017). Especially the underestimation of the biomarker-based $RH_{air}$
values under the 68% relative humidity conditions, as well as the large range in reconstructed $RH_{air}$
values for the 48, 49, 50% $RH_{air}$ chambers can be attributed to the leaf water reconstruction
uncertainties. It should be mentioned that using Eqs. 8 and 9 to calculate leaf water isotope
composition based on the biomarkers via a biosynthetic fractionation values implies that the
fractionation process in principle can be treated as single process with a unique source. While this
approximation can be questioned (see discussion in section 3.2), the overall approximation between
biomarker-based and measured $RH_{air}$ highlights the potential of the approach (Hepp et al., 2017;
Tuthorn et al., 2015; Zech et al., 2013a), also for future paleo-applications.

## 4 Conclusions
The climate chamber results and discussion suggest that leaf wax-derived $n$-alkane and hemicellulose-
derived sugar biomarkers are valuable $\delta^2H_{leaf-water}$ and $\delta^{18}O_{leaf-water}$ recorders, respectively. The coupling
of $\delta^2H_{n-alkane}$ and $\delta^{18}O_{sugar}$ results allows moreover a robust $RH_{air}$ reconstruction of the chambers in
which the plants were grown, by using simplified Craig-Gordon equations. With regard to the research
questions, we summarize as follows:

(i)  Alkanes with the chain-length $n$-C$_{29}$ were found to be suitable abundant for compound-
specific $\delta^2H$ measurements in the leaf samples from all investigated species (*Eucalyptus*
*globulus*, *Vicia faba* var. *minor* and *Brassica oleracea* var. *medullosa*). For *Vicia faba*,
additionally $n$-C$_{31}$ could be evaluated robustly. $\delta^{18}O_{sugar}$ values could be obtained for the
hemicellulose-derived monosaccharides arabinose and xylose.

(ii)  Both the $\delta^2H_{n-alkane}$ and $\delta^{18}O_{sugar}$ values yielded highly significant correlations with $\delta^2H_{leaf-water}$
and $\delta^{18}O_{leaf-water}$($r^2$ = 0.45 and 0.85, respectively; p < 0.001, n = 24). Mean fractionation
factors between biomarkers and leaf water were found to be -156‰ (ranging from -133
to - 192‰) for $\varepsilon_{n-alkane/leaf-water}$ and +27.3‰ (ranging from +23.0 to +32.3‰) for $\varepsilon_{sugar/leaf-water}$.

(iii)  Using measured leaf water isotope composition ($\delta^2H_{leaf-water}$ and $\delta^{18}O_{leaf-water}$) in a less (Eq.
6) and a more simplified rearranged Craig-Gordon model (Eq. 7), $RH_{air}$ and $RH_{leaf}$ can be
derived, by using either $T_{air}$ or $T_{leaf}$. Most accurately, the $RH_{air}$ values via Eq. 7 can be
reconstructed, with a calculated $R^2$ of 0.84 (p < 0.001) between measured and modeled
$RH_{air}$ and a RMSE of 6%. $RH_{leaf}$ reconstructions seemed less robust.

(iv)  Reconstructed source water isotope composition ($\delta^2H_s$, $\delta^{18}O_s$) are in range with the
measured tank water ($\delta^2H_{tank-water}$, $\delta^{18}O_{tank-water}$). However, modeled $\delta^2H_s$ and $\delta^{18}O_s$ show a
clear large range compared to $\delta^2H_{tank-water}$ and $\delta^{18}O_{tank-water}$. The uncertainties for source
water determination are thus considerably higher compared to the relative humidity
reconstructions. Still, the coupled $\delta^2H$-$\delta^{18}O$ approach enables a back calculation of the
plant source water. Uncertainties, with regard to relative humidity reconstructions via
biomarker-based leaf water isotope composition, arose from leaf water reconstructions
and model uncertainties, as shown in conclusions ii) and iii). Overall, the biomarker-based
and measured $RH_{air}$ correlation with a $R^2$ of 0.54 (p < 0.001) and a RMSE of 10% highlights
the great potential of the coupled $\delta^2H_{n-alkane}$-$\delta^{18}O_{sugar}$ paleohygrometer approach for
reliable relative humidity reconstructions.



## Acknowledgements

We would like to thank M. Bliedtner and J. Zech (both University of Bern) for help during lipid biomarker and $\delta^2H_{n\text{-alkane}}$ analysis. We thank M. Benesch (Martin-Luther-University Halle-Wittenberg) and M. Schaarschmidt (University of Bayreuth) for laboratory assistance during sugar biomarker and $\delta^{18}O_{sugar}$ analysis. The research was partly funded by the Swiss National Science Foundation (PP00P2 150590). We also acknowledge N. Orlowski (University of Freiburg), M. M. Lehmann (Swiss Federal Institute WSL, Birmensdorf) and L. Wüthrich (University of Bern) for helpful discussions. Involvement of K. Rozanski was supported by AGH UST statutory task No. 11.11.220.01/1 within subsidy of the Ministry of Science and Higher Education. J. Hepp greatly acknowledges the support given by the German Federal Environmental Foundation. The experiment carried out by C. Mayr was gratefully supported by the HGF-project "Natural climate variations from 10,000 years to the present" (project no. 01SF9813). The experiments were possible due to the assistance of J.B. Winkler, H. Lowag, D. Strube, A. Kruse, D. Arthofer, H. Seidlitz, D. Schneider, H. D. Payer, and other members of the Helmholtz Zentrum München.

## Author contributions

J. Hepp and M. Zech wrote the paper; C. Mayr was responsible for the climate chamber experiment together with W. Stichler and provided the leaf samples and the data; M. Zech and R. Zech were responsible for compound-specific isotope analysis on the biomarkers; J. Hepp, M. Tuthorn and I. K. Schäfer did laboratory work and data evaluation of the biomarker compound-specific isotope analysis; B. Glaser, D. Juchelka, K. Rozanski and all co-authors contributed to the discussion and commented on the manuscript.



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



## Figure captions

**Fig. 1:** A: Plant water (leaf water, xylem water and soil water) isotope compositions (in green, orange and brown, respectively) and the isotope composition of the investigated leaf biomarkers (leaf wax n-alkanes $n$-C$_{29}$ and $n$-C$_{31}$ as open diamonds and triangles, respectively; hemicellulose-derived sugars: arabinose and xylose as open squares and circles, respectively) for the three plants *Eucalyptus globulus, Vicia faba* and *Brassica oleracea* grown in the climate chambers. B: Associated climate chamber conditions (leaf temperature and relative humidity in green and air temperature and relative humidity in red). Error bars represent analytical standard deviation of the respective measurements (see section 2.2 and Mayr, 2002).

**Fig. 2:** $\delta^2$H-$\delta^{18}$O diagram illustrating the isotope composition of the biomarkers, comprising $\delta^2$H values of the leaf wax $n$-alkanes (C$_{29}$ for *Eucalyptus globulus* and *Brassica oleracea*; weighted mean of C$_{29}$ and C$_{31}$ for *Vicia faba*) and $\delta^{18}$O values of the hemicellulose-derived sugars arabinose and xylose (black crosses) and the measured isotope compositions of leaf water (green squares), xylem water (orange squares), soil water (brown squares), atmospheric water vapor (red squares) and the tank water used for irrigation (blue triangle), which plot very close to the global meteoric water line.

**Fig. 3:** Scatterplots depicting the relationships between the compound-specific biomarker isotope composition and the respective leaf water values (A: $\delta^2$H$_{n\text{-alkane}}$ vs. $\delta^2$H$_{\text{leaf-water}}$; B: $\delta^{18}$O$_{\text{sugar}}$ vs. $\delta^{18}$O$_{\text{leaf-water}}$). *Brassica oleracea, Eucalyptus globulus* and *Vicia faba* samples are shown in purple, orange and black, respectively. Error bars of the δ values represent standard deviation of repeated measurements (see section 2.2 and Mayr, 2002).

**Fig. 4:** Boxplots comprising the plant-specific fractionation between the biomarkers and the leaf water (A: ε$_{n\text{-alkane/leaf-water}}$ according Eq. 8; B: ε$_{\text{sugar/leaf-water}}$ according to Eq. 9). *Brassica oleracera, Eucalyptus globulus* and *Vicia faba* samples are shown in purple, orange and black, respectively. Boxplots show median (thick black line), interquartile range (IQR) with upper (75%) and lower (25%) quartiles, lower and upper whiskers, which are restricted to 1.5 · IQR. Outside the 1.5 · IQR space, the data points are marked with a dot. The notches are extend to ± 1.58 ·IQR/√n, by convention and give a 95% confidence interval for the difference of two medians (McGill et al., 1978).

**Fig. 5:** Scatterplots illustrating the correlation between leaf water-based and measured air/leaf relative humidity [modeled vs. measured RH$_{\text{air}}$ (A) and RH$_{\text{leaf}}$ (B)], modeled vs. measured leaf water deuterium-excess [T$_{\text{air}}$–based (B) and T$_{\text{leaf}}$-based (E) d$_e$ vs. deuterium-excess$_{\text{leaf-water}}$] and modeled vs. measured LEL slopes [T$_{\text{air}}$-based (C) and T$_{\text{leaf}}$-based (F) vs. measured slopes]. In red, the results of the less simplified models are displayed (Eq. 2 for d$_e$, Eq. 6 for RH and Eq. 10 for S$_{\text{LEL}}$) and in black the results of the more simplified models are shown (Eq. 3 and d$_e$, Eq. 7 for RH and Eq. 11 for S$_{\text{LEL}}$). Black lines indicate the 1:1 relationship. R$^2$ and RMSE are calculated as described in section 2.4, while the RMSE values have the dimensions of the respective variables. Error bars for the measured RH values represent analytical standard deviations (see Mayr, 2002). For the uncertainties of the calculated and modeled variables see section 2.4.

**Fig. 6:** Boxplots showing the measured leaf water in comparison to the biomarker-based leaf water (according Eqs. 8 and 9), tank water, source water calculated with biomarker-based leaf water values and source water based on measured leaf water. Source water isotope compositions were calculated via the slopes of the LEL´s (either with biomarker-based or measured leaf water values) and the GMWL.





The numbers (1-4) mark the available scenarios for source water reconstruction (see section 2.4): 1 = $S_{LEL}$ calculated according more simplified Eq. 11 with $T_{air}$, 2 = as 1 but with $T_{leaf}$, 3 = $S_{LEL}$ calculated according less simplified Eq. 10 with $T_{air}$, 4 = as 3 but with $T_{leaf}$. Boxplots show median (thick black line), interquartile range (IQR) with upper (75%) and lower (25%) quartiles, lower and upper whiskers, which are restricted to $1.5 \cdot$ IQR. Outside the $1.5 \cdot$ IQR space, the data points are marked with a dot. The notches are extend to $\pm 1.58 \cdot IQR/\sqrt{n}$, by convention and give a 95% confidence interval for the difference of two medians (McGill et al., 1978).

**Fig. 7:** Scatterplots depicting the relationship between biomarker-based (modeled) and measured air/leaf relative humidity [$RH_{air}$ (A) and $RH_{leaf}$ (B)]. Black lines indicate the 1:1 relationship. $R^2$ and RMSE was calculated as described in section 2.4, while the RMSE values have the dimensions of the respective variables. Error bars for the measured values represent analytical standard deviations (see Mayr, 2002). For uncertainty calculation of the modeled properties, see section 2.4. In addition, the leaf water-based air/leaf relative humidity results (from Fig. 5A and D) are shown in light colors for comparison.

**Fig. S1:** Boxplots comprising the plant-specific $\delta^2H_{n\text{-alkane}}$ (A) and $\delta^{18}O_{sugar}$ values (B). *Brassica oleracera, Eucalyptus globulus* and *Vicia faba* samples are shown in purple, orange and black, respectively. Boxplots show median (thick black line), interquartile range (IQR) with upper (75%) and lower (25%) quartiles, lower and upper whiskers, which are restricted to $1.5 \cdot$ IQR. Outside the $1.5 \cdot$ IQR space, the data points are marked with a dot. The notches are extend to $\pm 1.58 \cdot IQR/\sqrt{n}$, by convention and give a 95% confidence interval for the difference of two medians (McGill et al., 1978).

**Fig. S2:** Scatterplots of the fractionation between the biomarkers and leaf water vs. air temperature, air relative humidity (A and B: $\varepsilon_{n\text{-alkane/leaf-water}}$ according Eq. 13; C and D $\varepsilon_{sugar/leaf-water}$ according Eq. 14). *Brassica oleracera, Eucalyptus globulus* and *Vicia faba* samples are shown in purple, orange and black, respectively. Error bars for the measured values represent analytical standard deviations of repeated measurements (see section 2.2 and Mayr, 2002). For uncertainty calculation of the $\varepsilon$ values, see section 2.4.

**Fig. S3:** Boxplots comprising measured and modeled RH (A) and deuterium-excess values (B). The numbers (1-2) mark the two available models for $RH_{leaf/air}$ and $d_e$ reconstruction (see section 2.4): 1 = more simplified models (Eq. 3 for $d_e$ and Eq. 7 for RH), 2 = less simplified models (Eq. 2 for $d_e$ and Eq. 6 for RH). Boxplots show median (thick black line), interquartile range (IQR) with upper (75%) and lower (25%) quartiles, lower and upper whiskers, which are restricted to $1.5 \cdot$ IQR. Outside the $1.5 \cdot$ IQR space, the data points are marked with a dot. The notches are extend to $\pm 1.58 \cdot IQR/\sqrt{n}$, by convention and give a 95% confidence interval for the difference of two medians (McGill et al., 1978).





**Fig. 1**

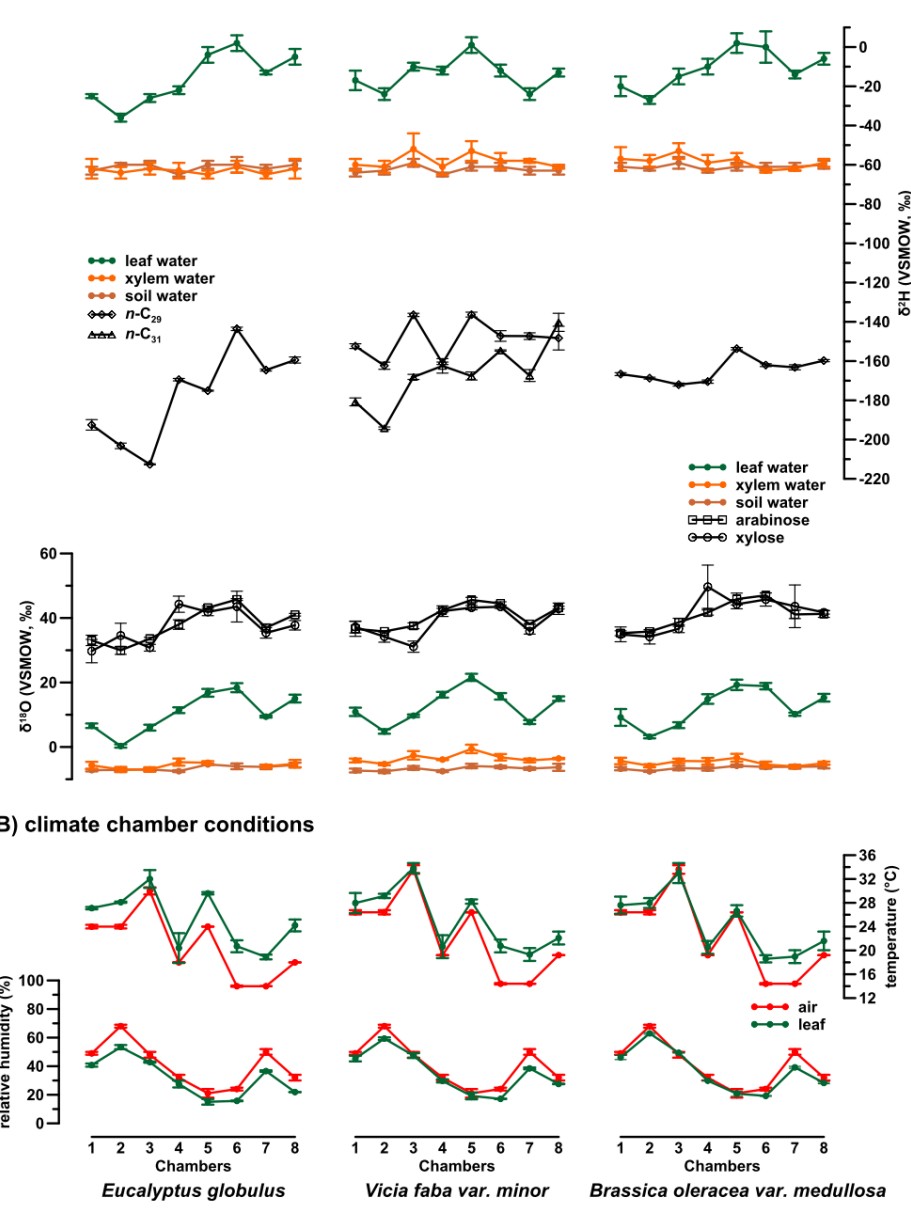






**Fig. 2**

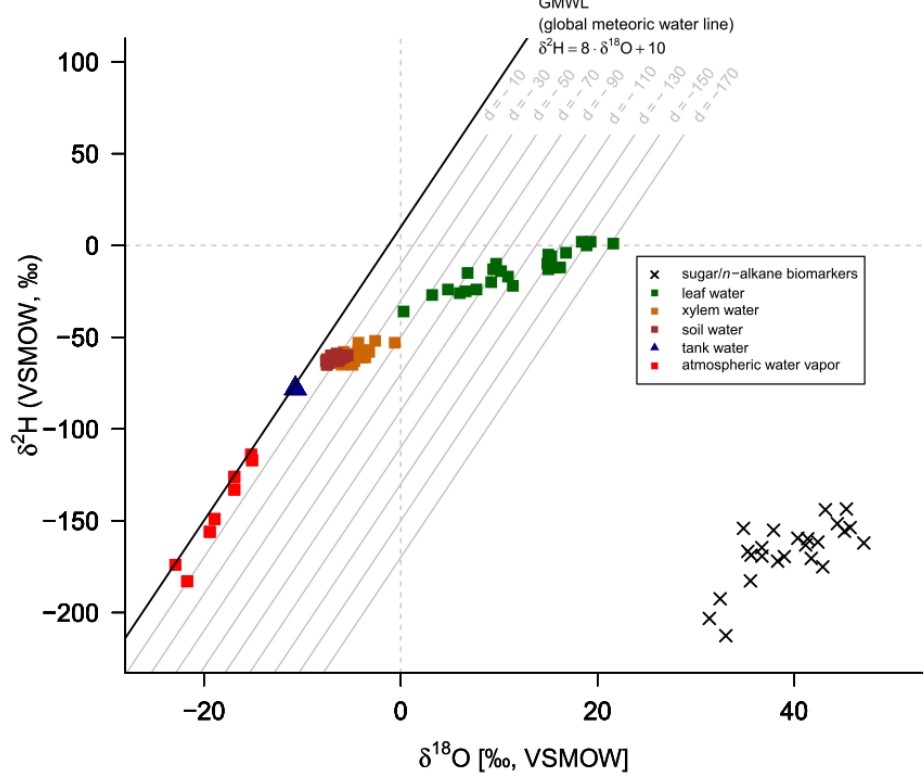






**Fig. 3**

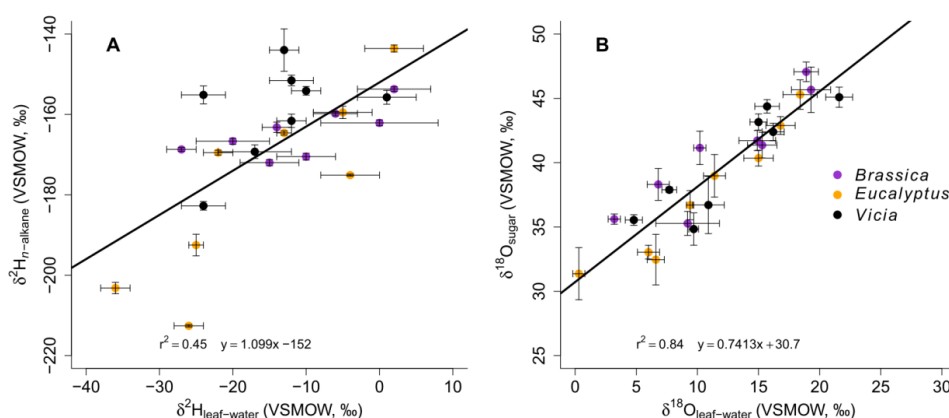

**Fig. 4**

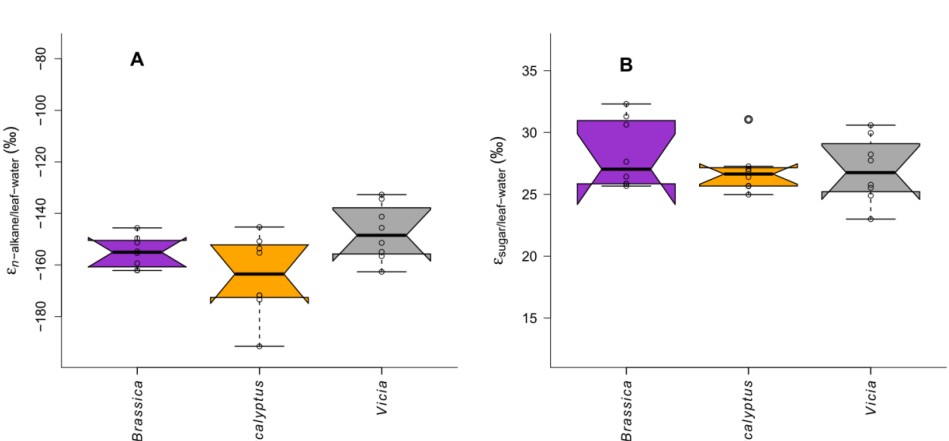






**Fig. 5**








**Fig. 6**

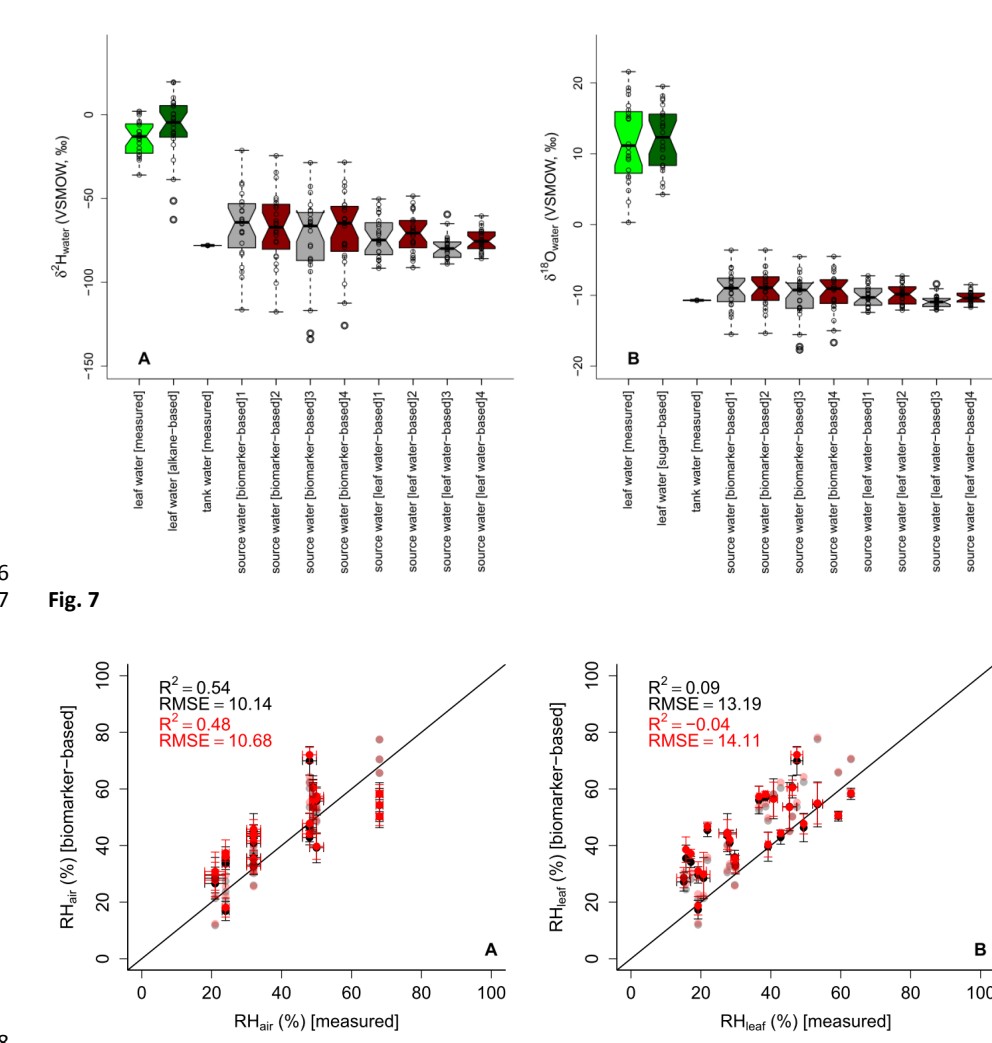

**Fig. 7**





**Fig. S1**

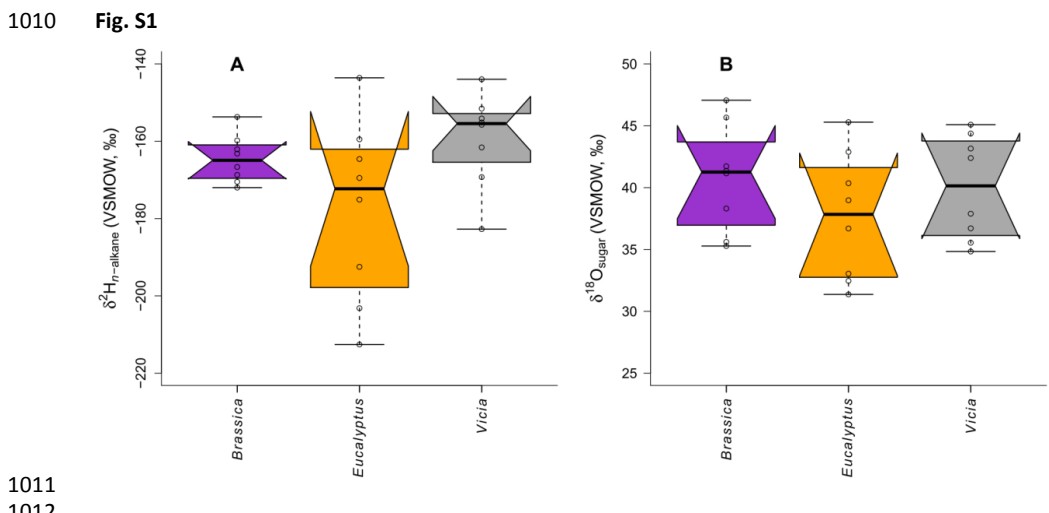






**Fig. S2**



**Fig. S3**