# Peer review of "approach based on a climate chamber experiment"

_Biogeosciences, 2019_

## Referee Comment (RC1) · Anonymous Referee #1 · 21 Nov 2019

BG-2019-427

Dear Johannes Hepp and co-authors,

I have to say that I really like you combined water isotope method for reconstructing source water isotope values and relative humidity. I found this manuscript extremely difficult to read, however. It is quite complex and difficult to keep track of all the different measured and modeled parameters and the writing and structure doesn't make this easier. I'll give a few examples of this in the more detailed comments. I do think if you significantly clean up the manuscript structure and writing issues it might be a worthwhile contribution to Biogeosciences, as is it is a bit difficult to judge. I do have

a few, more general, comments. You have the tendency to position your combined method opposite to the n-alkane based reconstructions, by writing things like "However, a direct reconstruction of the isotopic composition of paleoprecipitation based on $\delta$2Hn-alkane alone can be challenging due to the overprint of the source water isotopic signal by leaf-water enrichment". The thing is the same holds true for $\delta$18Ocellulose/sugars , so this is not necessarily a "bad" thing of the n-alkane method, in fact for reconstructing RH it is a good thing there is leaf-water enrichment, otherwise there would be nothing to reconstruct. Personally, I think all methods/proxies have their issues and we need to know what they are so we can work with them. Measuring both hydrogen and oxygen isotopes has its advantages, I can see that, not only do you get duplication, the combination of the two opens up new possibilities as well. For this you basically work back to the global meteoric waterline, is that always the most appropriate meteoric waterline to use? As far as I know local meteoric waterlines can differ quite a bit from the global, right? And also with the use as paleo proxy, meteoric waterlines might not be constant over time, glacial/interglacial changes for instance? Would that complicate that the use of your proxy and how? Do the field studies look at global or local meteoric waterlines, for instance? My last question deals with exchangeable oxygen as mentioned on page 11, "The relatively uniform fractionation is explained via the isotope exchange between the carbonyl oxygens of the organic molecules and the surrounding water (cf. Schmidt et al., 2001)". You seem to suggest this might play a role in your sample set, right? But your samples are from 2001, so what would be the water is exchanged with, the water from the experimental set-up (leaf or other), from the freezer it was stored in or the lab they were analyzed in? Or perhaps oxygen form the organic solvents used? How would this affect your method, if at all, and calculating back to the global meteoric waterline, for instance?

Detailed comments:

Line 54: the leaf water data has been published already right? In Mayr 2002, so what is new here? The modeling? Line 55-56 and throughout the manuscript: I find the use

of the more simplified or the less simplified C-G model very confusing. In the abstract it is even a bit meaningless since the reader has no idea what you mean. Perhaps you can give them a different name? First paragraph of the introduction: Be careful with taking down the n-alkane method, you need this leaf water enrichment of both H and O for your reconstructions. At the end of this paragraph I expect how your method will solve these issues, but it is not there. Second paragraph of the introduction: I think there are publications on isotope fractionation during water uptake through the roots, use them. There is also information on soil enrichment due to evaporation, it might have been discarded as major factor in most cases, but still interesting to mention, I think. Line 104: It is known to not be constant! You need to address this at least a little. 2.1: What about the return flow of enriched leaf water into the stems? How might that affect your measurements? You measured both xylem and phloem and called it xylem. Line 170: high temperature conversion (HT-EA-irMS) Line 179/200: use ‰ and not per mil. Line 181: strange definition of the delta notation. Line 258: less simplified results? The results of the less simplified (more complex or more correct) C-G model? Line 299: this is a bit worrying, the low number of samples per plant species prohibits a robust interpretation. . ... Line 328: so fractionation is not stable. . . introduce this in the introduction. . . Line 338: Epsilonbio is large and not constant and one of the main reasons for this NADPH and the sources of H for NADPH and the pathways by which NADP+ is reduced to NADPH, the enzymes involved etc. etc. It is not just another additional complicating factor. Line 359: What is this unenriched source water? Source water as in "precipitation" or in the experiment the big tank of water sourcing everything? Or source water as in the source for biosynthesis, so water in the leaf that has not been enriched by evaporation? Wouldn't that mix or is it compartmentalized? How does what water end up where, this is confusing. Line 370: Now it is even depleted source water contributing to local synthesis water? How did it get depleted or is it depleted relative to the enriched leaf water? The same as the non-enriched source water? And the source is in this case the source for biosynthesis? Righ? Line 384: I thought I read -152 earlier in the manuscript as average? Line 403: This range from -

192 to -133 was for Epsilonn-alkane-leaf water on the previous page? So do you mean Epsilonxylem water-leaf water here? Same for line 418? Already difficult to keep track of everything. Line 418: NADPH, photosynthesis, pentose phosphate pathway, etc. etc. etc. . ... Line 428-429: Oh, ok. . .. See above. Line 478: despite without? Which can be expected from literature? The latter definitely needs references, but I lost track of the meaning of this sentence. Line 484-485: an overestimation of models? Of the quality of the models, the outcome of the models? Line 488: There it is back diffusion of enriched waters? Did that affect your stem water measurements? Line 490-491: difficult to understand, there is no obvious reason for it? Line 494-495: measured values based on the less simplified model? Are they measured or modeled? The more complex or more correct model. . . Line 499-502: I am lost here. The more simplified models are still more accurate, despite the less simplified models do not reflect well the range of the measured Slel. Much better matches are found for the less simplified LEL slopes. . ... So the less simplified models are both worse and better? It's unclear to me, so please try to rewrite in way that is easier to understand. Line 535: the overlapping notches are a graphical representation of the lack of a statistical difference, right? ID so, don't use "with regard", but something like as illustrated by or so. Line 566: . ..., can be explained as for the leaf water based application. Again I am lost here.

Again, I am always in favor of new proxies being developed and this an interesting development. I do however think that the manuscript as is will not do this new proxy any favors. This manuscript could benefit greatly from some restructuring and rewriting. The data deserves that.

---

## Referee Comment (RC2) · Anonymous Referee #2 · 2 Dec 2019

The manuscript written by Hepp and colleagues presents results of laboratory experiments where different plant species were grown and hydrogen and oxygen isotope ratios were measured on different organic compounds. The dual isotope approach is a valuable and important step toward better paleoclimate reconstructions, but I wonder how comparable these two compounds are. There are differences in the ways these two compounds are synthesized and I think a more in depth discussion of these mechanisms is necessary in order to confidently use them, especially for paleoenvironmental reconstructions.

Overall, the manuscript is rather lengthy and could be made more concise by refocus-

ing the discussion. The discussions about biosynthesis should be revisited and revised, because as written now they are a bit unclear. It might be worth it to discuss biosynthesis and effects that might have on isotopic values first, then move to a discussion about how comparable isotopic values of these two compounds really are. This could be followed by extracellular factors that influence these proxies and the comparison with published data and what this might mean overall. There is also a model presented here, but the results of that model are peppered throughout the discussion which make it difficult to follow. It would be good to make this clear, perhaps by dedicating a section solely to the model-data comparison. Finally, a number of sentences would benefit from restructuring because as written now they are hard to follow. Please pay attention to grammar and appropriate phrasing throughout.

Specific comments:

Lines 42-44 : Consider rewording this to: 'can relative humidity be accurately reconstructed from leaf water isotope values'.

Line 43: Should be 'enable'

Line 45: robust source water reconstruction?

Line 60: it might be better to explain this differently. 'getting worse' sounds very informal.

Line 73: 'with respect to' instead of 'in respect'

Line 80: It would be good to discuss the correlation between d2H and d18O in meteoric waters here.

Line 82: Please explain the climate transect. Altitudinal ?

Lines 123-124: were these temperature and humidity values for all of the chambers? Please better explain the set up, e.g., two chambers were kept at a temperature of X and humidity of Y. Also, please remove the additional 'and' on line 124.

Line 152: pyrolysis mode

Line 211: 'where' not 'were'

Line 290: weighted mean of C29 and C31 ?

Line 314: why is it better to use the weighted mean instead of the individual d18O for arabinose and xylose ?

Line 322: what is the offset?

Line 328: change 'relation' to 'correlation'

Lines 407 – 412 : The way you discuss the biosynthesis here is unclear. It reads like you are saying hydrogen is added to a lipid in the chloroplast and the cytosol and on top of that photosynthesis and the pentose phosphate pathway add other hydrogen. NADPH is reduced by different sources in the chloroplast and the cytosol (see Schmidt et al., 2003). This reduced NADPH is then used in lipid biosynthesis in these separate compartments. Please be careful how you discuss this. Also it should be pentose phosphate 'pathway' not cycle. Furthermore, are you sure the n-alkanes are synthesized in the cytosol and not in the endoplasmic reticulum? The Schmidt et al. (2003) and Cormier et al. (2018) papers both provide excellent explanations of this and effects of biosynthesis on isotopic fractionation of lipids (specifically have a look at figure 5 from Cormier et al., 2018 for the n-alkane synthesis). Finally, on line 408: 'modifying/expanding fatty acids' should be changed to 'elongation of fatty acids'.

Figure 1A: It is difficult to distinguish the different shapes in this figure. It might be helpful to remove the lines from these plots. The colors from xylem water and soil water are very similar. You might consider choosing two colors with more contrast.

[Figure]

---

## Author Comment (AC1) · 21 Jan 2020

**Reply to Referee #1**

by Johannes Hepp, Michael Zech & co-authors

*Dear Johannes Hepp and co-authors, I have to say that I really like you combined water isotope method for reconstructing source water isotope values and relative humidity. I found this manuscript extremely difficult to read, however. It is quite complex and difficult to keep track of all the different measured and modeled parameters and the writing and structure doesn't make this easier. I'll give a few examples of this in the more detailed comments. I do think if you significantly clean up the manuscript structure and writing issues it might be a worthwhile contribution to Biogeosciences, as is it is a bit difficult to judge. I do have a few, more general, comments.*

→ We are very grateful to anonymous Referee #1 for her/his encouraging words concerning our coupled $\delta^2H_{n\text{-alkane}}$- $\delta^{18}O_{sugar}$ paleohygrometer approach. We also agree that the original MS is quite complex and difficult to read; that's why we are furthermore very grateful to Referee #1 for her/his constructive specific suggestions helping to improve our manuscript and we will be happy to include her/him in the acknowledgements. Please find our replies to the individual comments below.

*You have the tendency to position your combined method opposite to the n-alkane based reconstructions, by writing things like "However, a direct reconstruction of the isotopic composition of paleoprecipitation based on δ2Hn-alkane alone can be challenging due to the overprint of the source water isotopic signal by leaf-water enrichment". The thing is the same holds true for δ18Ocellulose/sugars, so this is not necessarily a "bad" thing of the n-alkane method, in fact for reconstructing RH it is a good thing there is leaf-water enrichment, otherwise there would be nothing to reconstruct. Personally, I think all methods/proxies have their issues and we need to know what they are so we can work with them. Measuring both hydrogen and oxygen isotopes has its advantages, I can see that, not only do you get duplication, the combination of the two opens up new possibilities as well. For this you basically work back to the global meteoric waterline, is that always the most appropriate meteoric waterline to use? As far as I know local meteoric waterlines can differ quite a bit from the global, right? And also with the use as paleo proxy, meteoric waterlines might not be constant over time, glacial/interglacial changes for instance? Would that complicate that the use of your proxy and how? Do the field studies look at global or local meteoric waterlines, for instance?*

→ Thank you for pointing us to avoid possibly offending formulations. We will readily rewrite the referred sentence and in addition check the rest of our MS for respective formulations.

→ We fully agree with Reviewer #1 that in many paleoclimate applications the usage of a local meteoric water line (LMWL) is preferable over the global meteoric water line (GMWL). We also agree, that neither the GMWL nor the LMWLs are/were constant over time. At least neither for Greenland and Antarctic ice cores (Masson-Delmotte et al., 2005; Stenni et al., 2010) nor for Central Europe (Rozanski, 1985) the d-excess, which is relevant for our approach, does/did

however, exceed roughly 4 ‰. Given that moreover the tank water used in our climate chamber experiment plots close to the GMWL, we consider the GMWL to be appropriate for our case study.

*My last question deals with exchangeable oxygen as mentioned on page 11, "The relatively uniform fractionation is explained via the isotope exchange between the carbonyl oxygens of the organic molecules and the surrounding water (cf. Schmidt et al., 2001)". You seem to suggest this might play a role in your sample set, right? But your samples are from 2001, so what would be the water is exchanged with, the water from the experimental set-up (leaf or other), from the freezer it was stored in or the lab they were analyzed in? Or perhaps oxygen form the organic solvents used? How would this affect your method, if at all, and calculating back to the global meteoric waterline, for instance?*

→ Yes, oxygen exchange is indeed, as raised by Reviewer #1, an important issue. According to our current state of knowledge, oxygen exchange from organic solvents or during sample storage can be excluded (O in hydroxyl groups of sugar biomarkers is not exchangeable, cf. Zech et al., 2012). By contrast, the oxygen atoms in C1 position of the sugar biomarkers are introduced during the hydrolytic cleavage of the sugar monomers from the hemicelluloses and need to be corrected for mathematically (Zech and Glaser, 2009). We will include a respective sentence in the revised manuscript.

*Detailed comments:*

*Line 54: the leaf water data has been published already right? In Mayr 2002, so what is new here? The modeling?*

→ Yes, you are right. Leaf water isotope values were already published by Mayr (2002) and Zech et al. (2014), but modelling/calculation/reconstruction of relative humidity values from those data is firstly presented here.

*Line 55-56 and throughout the manuscript: I find the use of the more simplified or the less simplified C-G model very confusing. In the abstract it is even a bit meaningless since the reader has no idea what you mean. Perhaps you can give them a different name?*

→ We agree with Reviewer#1 that the use of a "more" or "less" simplified CG-model is confusing. During revision, we will therefore refrain from this differentiation and include a short respective explanation or footnote instead.

*First paragraph of the introduction: Be careful with taking down the n-alkane method, you need this leaf water enrichment of both H and O for your reconstructions. At the end of this paragraph I expect how your method will solve these issues, but it is not there.*

→ Thanks. We will readily change the formulation and rewrite the end of the paragraph.

*Second paragraph of the introduction: I think there are publications on isotope fractionation during water uptake through the roots, use them. There is also information on soil enrichment due to evaporation, it might have been discarded as major factor in most cases, but still interesting to mention, I think.*

→ You are right. We will update the citations in this paragraph and also mention and discuss the factor of soil water enrichment during the revision of the manuscript.

*Line 104: It is known to not be constant! You need to address this at least a little.*

→ You are right. We will readily change this sentence during the revision of the manuscript.

*2.1: What about the return flow of enriched leaf water into the stems? How might that affect your measurements? You measured both xylem and phloem and called it xylem.*

→ The so called "Peclet-Effect" was indeed discussed extensively in the literature (see e.g. the review of Cernusak et al., 2016). This effect does very likely also affect the presented "xylem water" dataset. That's why we explicitly state that "stem water is a mixture of xylem and phloem water". During the revision of the manuscript we will carefully check if the presentation of this data is really needed (see also reply from above).

*Line 170: high temperature conversion (HT-EA-irMS) Line 179/200: use ‰ and not per mil.*

→ Will be changed in the revised manuscript.

*Line 181: strange definition of the delta notation.*

→ We will update the definition according the current literature.

*Line 258: less simplified results? The results of the less simplified (more complex or more correct) C-G model?*

→ As mentioned in our reply above, we will readily simplify the MS during revision and refrain from using "more" or "less" simplified GC models.

*Line 299: this is a bit worrying, the low number of samples per plant species prohibits a robust interpretation.....*

→ We will rewrite this sentence during revision because we wanted to say that a more statistically verified statement is not possible due to the relative low number of samples per species. While we are aware that the low number of samples introduces some uncertainty, we are confident that our overall data evaluation and interpretation is indeed robust.

*Line 328: so fractionation is not stable... introduce this in the introduction...*

→ As mentioned above, we will readily change this during the revision of the manuscript.

*Line 338: Epsilonbio is large and not constant and one of the main reasons for this NADPH and the sources of H for NADPH and the pathways by which NADP+ is reduced to NADPH, the enzymes involved etc. etc. It is not just another additional complicating factor.*

→ You are right. During the revision of the manuscript, we will also rewrite this part of the discussion.

*Line 359: What is this unenriched source water? Source water as in "precipitation" or in the experiment the big tank of water sourcing everything? Or source water as in the source for biosynthesis, so water in the leaf that has not been enriched by evaporation? Wouldn't that mix or is it compartmentalized? How does what water end up where, this is confusing.*

→ Again, we apologize for the unclear way of writing. Here in this specific case we use "source water" *sensu* water being used for biosynthesis. During revision of the manuscript we will do our best to avoid such confusing formulations.

*Line 370: Now it is even depleted source water contributing to local synthesis water? How did it get depleted or is it depleted relative to the enriched leaf water? The same as the non-enriched source water? And the source is in this case the source for biosynthesis? Right?*

→ It is depleted compared to the enriched leaf water and therefore called non-enriched source water. As mentioned in our reply above, we will carefully check the whole manuscript for confusing formulations.

*Line 384: I thought I read -152 earlier in the manuscript as average?*

→ -152‰ is the intercept of the $\delta^2 H_{n\text{-alkane}}$ to $\delta^2 H_{\text{leaf-water}}$ relationship. By contrast, the mean of the calculated $\varepsilon_{n\text{-alkane/leaf-water}}$ is -156‰.

*Line 403: This range from -192 to -133 was for Epsilonn-alkane-leaf water on the previous page? So do you mean Epsilonxylem water-leaf water here? Same for line 418? Already difficult to keep track of everything.*

→ Please accept our apologies for this confusion. In line 403 and 418 we meant, as in the whole paragraph, $\varepsilon_{n\text{-alkane/leaf-water}}$ and not the fractionation between leaf water and xylem water. We will do our best to simplify formulations during revision.

*Line 418: NADPH, photosynthesis, pentose phosphate pathway, etc. etc. etc....*

→ You are right. In order to simplify our MS, we will likely delete these sentences during revision.

*Line 428-429: Oh, ok.... See above.*

→ We acknowledge that the fractionation between the biomarkers and the leaf water is not constant. As mentioned above, we will state this more clearly during revision.

*Line 478: despite without? Which can be expected from literature? The latter definitely needs references, but I lost track of the meaning of this sentence.*

*Line 484-485: an overestimation of models? Of the quality of the models, the outcome of the models?*

→ Sorry, we will readily rewrite these sentences during revision and include references.

*Line 488: There it is back diffusion of enriched waters? Did that affect your stem water measurements?*

→ Yes, you are right, there is very likely a partial influence by back diffusion. Given that stem/xylem water is not crucial for our data evaluation/interpretation, we will likely refrain from presenting and discussing these data during revision.

*Line 490-491: difficult to understand, there is no obvious reason for it?*

→ We wanted to state with this sentence that we do not have a proper explanation why the modeled deuterium-excess values do not show a positive offset compared to the measured values. Will be clarified/rewritten during revision.

*Line 494-495: measured values based on the less simplified model? Are they measured or modeled? The more complex or more correct model...*

→ We apology for this unclear sentence. It is meant that the more complex model produces a much better $S_{LEL}$ outcome that the less complex model, however, without producing better model outcome regarding RH and deuterium-excess. We will clarify this during revision.

*Line 499-502: I am lost here. The more simplified models are still more accurate, despite the less simplified models do not reflect well the range of the measured Slel. Much better matches are found for the less simplified LEL slopes.... So the less simplified models are both worse and better? It's unclear to me, so please try to rewrite in way that is easier to understand.*

→ Again we apology for this unclear style of writing. As mentioned above, we will do our best to clarify/improve this during revision.

*Line 535: the overlapping notches are a graphical representation of the lack of a statistical difference, right? ID so, don't use "with regard", but something like as illustrated by or so.*

→ You are right. We will readily change this.

*Line 566: ...., can be explained as for the leaf water based application. Again, I am lost here. Again, I am always in favor of new proxies being developed and this an interesting development. I do however think that the manuscript as is will not do this new proxy any favors. This manuscript could benefit greatly from some restructuring and rewriting. The data deserves that.*

→ We fully agree with Reviewer#1 and are very grateful for her/his support. We will do our best to simplify, restructure and rewrite our MS during the revision.

**References**

Cernusak, L. A., Barbour, M. M., Arndt, S. K., Cheesman, A. W., English, N. B., Feild, T. S., Helliker, B. R., Holloway-Phillips, M. M., Holtum, J. A. M., Kahmen, A., Mcinerney, F. A., Munksgaard, N. C., Simonin, K. A., Song, X., Stuart-Williams, H., West, J. B. and Farquhar, G. D.: Stable isotopes in leaf water of terrestrial plants, Plant Cell and Environment, 39(5), 1087–1102, doi:10.1111/pce.12703, 2016.

Masson-Delmotte, V., Jouzel, J., Landais, A., Stievenard, M., Johnson, S. J., White, J. W. C., Werner, M., Sveinbjornsdottir, A. and Fuhrer, K.: GRIP Deuterium Excess Reveals Rapid and Orbital-Scale Changes in Greenland Moisture Origin, Science, 309, 118–121, doi:10.1126/science.1108575, 2005.

Mayr, C.: Möglichkeiten der Klimarekonstruktion im Holozän mit $\delta^{13}C$- und $\delta^{2}H$-Werten von Baum-Jahrringen auf der Basis von Klimakammerversuchen und Rezentstudien, PhD thesis, Ludwig-Maximilians-Universität München. GSF-Bericht 14/02, 152 pp., 2002.

Stenni, B., Masson-Delmotte, V., Selmo, E., Oerter, H., Meyer, H., Röthlisberger, R., Jouzel, J.,

Cattani, O., Falourd, S., Fischer, H., Hoffmann, G., Iacumin, P., Johnsen, S. J., Minster, B. and Udisti, R.: The deuterium excess records of EPICA Dome C and Dronning Maud Land ice cores (East Antarctica), Quaternary Science Reviews, 29, 146–159, doi:10.1016/j.quascirev.2009.10.009, 2010.

Zech, M. and Glaser, B.: Compound-specific $\delta^{18}O$ analyses of neutral sugars in soils using gas chromatography-pyrolysis-isotope ratio mass spectrometry: problems, possible solutions and a first application, Rapid Communications in Mass Spectrometry, 23, 3522–3532, doi:10.1002/rcm, 2009.

Zech, M., Werner, R. A., Juchelka, D., Kalbitz, K., Buggle, B. and Glaser, B.: Absence of oxygen isotope fractionation/exchange of (hemi-) cellulose derived sugars during litter decomposition, Organic Geochemistry, 42(12), 1470–1475, doi:http://dx.doi.org/10.1016/j.orggeochem.2011.06.006, 2012.

Zech, M., Mayr, C., Tuthorn, M., Leiber-Sauheitl, K. and Glaser, B.: Oxygen isotope ratios ($^{18}O/^{16}O$) of hemicellulose-derived sugar biomarkers in plants, soils and sediments as paleoclimate proxy I: Insight from a climate chamber experiment, Geochimica et Cosmochimica Acta, 126(0), 614–623, doi:http://dx.doi.org/10.1016/j.gca.2013.10.048, 2014.

---

## Author Comment (AC2) · 21 Jan 2020

**Reply to Referee #2**

by Johannes Hepp, Michael Zech & co-authors

*The manuscript written by Hepp and colleagues presents results of laboratory experiments where different plant species were grown and hydrogen and oxygen isotope ratios were measured on different organic compounds. The dual isotope approach is a valuable and important step toward better paleoclimate reconstructions, but I wonder how comparable these two compounds are. There are differences in the ways these two compounds are synthesized and I think a more in depth discussion of these mechanisms is necessary in order to confidently use them, especially for paleoenvironmental reconstructions.*

→ We are very grateful to anonymous Referee #2 for her/his judgement that our coupled $\delta^2H_{n\text{-}alkane}$- $\delta^{18}O_{sugar}$ paleohygrometer approach is a valuable and important step towards better paleoclimate reconstructions. We agree that *n*-alkane and sugar biomarkers are biosynthesized differently. However, the principles are the same for both biomarker classes: they (i) reflect the isotopic composition of precipitation modified by (ii) leaf water enrichment due to evapotranspiration and (iii) biosynthetic fractionation. We feel that a more in depth discussion of further specific and complicating mechanistic details would be beyond the scope of this manuscript. As pointed out by Reviewer#2 (and Reviewer#1), our MS is already rather lengthy and needs refocusing. For further mechanistic details we therefore suggest to refer our readers to the literature.

*Overall, the manuscript is rather lengthy and could be made more concise by refocusing the discussion. The discussions about biosynthesis should be revisited and revised, because as written now they are a bit unclear. It might be worth it to discuss biosynthesis and effects that might have on isotopic values first, then move to a discussion about how comparable isotopic values of these two compounds really are. This could be followed by extracellular factors that influence these proxies and the comparison with published data and what this might mean overall. There is also a model presented here, but the results of that model are peppered throughout the discussion which make it difficult to follow. It would be good to make this clear, perhaps by dedicating a section solely to the model-data comparison. Finally, a number of sentences would benefit from restructuring because as written now they are hard to follow. Please pay attention to grammar and appropriate phrasing throughout.*

→ Thank you for raising these issues and providing suggestions how to restructure and improve our manuscript. We will do our best to improve clarity during the revision.

*Specific comments:*

*Lines 42-44 : Consider rewording this to: 'can relative humidity be accurately reconstructed from leaf water isotope values'.*

→ We will readily change this as suggested.

*Line 43: Should be 'enable'*

→ Will be changed.

*Line 45: robust source water reconstruction?*

→ As mentioned in the replies to Referee #1, we will readily clarify the whole manuscript also regarding source water (e.g. plant source water vs. source water for biosynthesis…).

*Line 60: it might be better to explain this differently. 'getting worse' sounds very informal.*

→ Will be changed.

*Line 73: 'with respect to' instead of 'in respect'*

→ Will be changed.

*Line 80: It would be good to discuss the correlation between d2H and d18O in meteoric waters here.*

→ Will be added.

*Line 82: Please explain the climate transect. Altitudinal?*

→ This information will be added.

*Lines 123-124: were these temperature and humidity values for all of the chambers? Please better explain the set up, e.g., two chambers were kept at a temperature of X and humidity of Y. Also, please remove the additional 'and' on line 124.*

→ Will be changed.

*Line 152: pyrolysis mode*

→ Will be changed.

*Line 211: 'where' not 'were'*

→ Will be changed.

*Line 290: weighted mean of C29 and C31?*

→ Yes, will be added.

*Line 314: why is it better to use the weighted mean instead of the individual d18O for arabinose and xylose?*

→ We expect the individual sugars to be more prone to analytical uncertainties. Please compare with alkanes, where often weighted means are used, too.

*Line 322: what is the offset?*

→ The offset reflects the fractionation between biomarker and leaf water ($\varepsilon_{bio}$). This will be added during the revision.

*Line 328: change 'relation' to 'correlation'*

→ Will be changed.

*Lines 407 – 412 : The way you discuss the biosynthesis here is unclear. It reads like you are saying hydrogen is added to a lipid in the chloroplast and the cytosol and on top of that photosynthesis and the pentose phosphate pathway add other hydrogen. NADPH is reduced by different sources in the chloroplast and the cytosol (see Schmidt et al., 2003). This reduced NADPH is then used in lipid biosynthesis in these separate compartments. Please be careful how you discuss this. Also it should be pentose phosphate 'pathway' not cycle. Furthermore, are you sure the n-alkanes are synthesized in the cytosol and not in the endoplasmic reticulum? The Schmidt et al. (2003) and Cormier et al. (2018) papers both provide excellent explanations of this and effects of biosynthesis on isotopic fractionation of lipids (specifically have a look at figure 5 from Cormier et al., 2018 for the n-alkane synthesis). Finally, on line 408: 'modifying/expanding fatty acids' should be changed to 'elongation of fatty acids'.*

→ Thank you for raising this issue and giving these explanations. Will be checked and changed.

*Figure 1A: It is difficult to distinguish the different shapes in this figure. It might be helpful to remove the lines from these plots. The colors from xylem water and soil water are very similar. You might consider choosing two colors with more contrast.*

→ Will be changed.